# Towards Learning POMDPs Without Full Observability

## Abstract

We are interested in enabling autonomous agents to learn and reason about systems with hidden states, such as furniture with hidden locking mechanisms. We cast this problem as learning the parameters of a discrete Partially Observable Markov Decision Process (POMDP). The agent begins with knowledge of the POMDP's actions and observation spaces, but not its state space, transitions, or observation models. These properties must be constructed from action-observation sequences. Spectral approaches to learning models of partially observable domains, such as learning Predictive State Representations (PSRs), are known to directly estimate the number of hidden states. These methods cannot, however, yield direct estimates of transition and observation likelihoods, which are important for many downstream reasoning tasks. Other approaches leverage tensor decompositions to estimate transition and observation likelihoods but often assume full state observability and full-rank transition matrices for all actions. To relax these assumptions, we study how PSRs learn transition and observation matrices up to a similarity transform, which may be estimated via tensor methods. Our method learns observation matrices and transition matrices up to a partition of states, where the states in a single partition have the same observation distributions corresponding to actions whose transition matrices are full-rank. Our experiments suggest that these partition-level transition models learned by our method, with a sufficient amount of data, meets the performance of PSRs as models to be used by standard sampling-based POMDP solvers. Furthermore, the explicit observation and transition likelihoods can be leveraged to specify planner behavior after the model has been learned.

## 1 Introduction

When planning and acting in the real world, intelligent agents must learn and reason about hidden information. Of great inspiration to us is the work of Baum et al. (2017), which shows that a real autonomous robot can infer a cabinet's locking mechanism from a hypothesis set of possible mechanisms through interaction. We are interested in a more general problem where autonomous agents must learn, through interaction, the dynamics of a system with hidden states, without any knowledge of the system state and transitions beforehand. The agent should also compute explicit estimates of transition and observation likelihoods to support downstream operations that manipulate the model, such as the specification of tasks to direct agent behavior. Our problem is modeled as learning the parameters of a discrete Partially Observable Markov Decision Process (POMDP) from a sequence of actions and observations acquired through random exploration.

One common approach to learning a representation of a probabilistic latent-variable models like a POMDP is to apply a spectral decomposition to a matrix that contains estimates of the joint likelihoods of the observable random variables (Hsu et al., 2012; Balle et al., 2014). In particular, singular-value decompositions (SVD) give a way of estimating the number of hidden variables of the system by truncating the singular values under a certain threshold. For POMDPs, spectral methods may be applied to a *Hankel matrix*, which stores the joint likelihood between past and future observations conditioned on a sequence of past and future actions. The decomposition of this matrix can be used to derive a (linear) Predictive State Representation, which can be interpreted as an automaton with real-valued transition matrices (Boots et al., 2011; Balle et al., 2014). The 'state' of the PSR is a sufficient statistic that can be used to predict the likelihood of future observations given a

possible sequence of actions. This prediction capability allows PSRs to be used as black-box models for reinforcement learning (Liu et al., 2022; Zhan et al., 2022), but since transition and observation likelihoods cannot be directly read from a PSR, these models are difficult to manipulate after they are learned.

There are other POMDP-learning algorithms that yield estimates of observation and transition likelihoods, but under assumptions that ultimately restrict the class of POMDPs that can be learned. Approaches introduced by Azizzadenesheli et al. (2016) and Guo et al. (2016) utilize tensor decompositions to recover observation distributions for each action whose transition matrix is full-rank. To recover the transitions, however, these approaches must also make the assumption that for each action, the corresponding observation distribution must be unique for every state. Full-rank transitions are common when modeling many real-world POMDPs, especially when actions may 'fail' with some probability, causing the system state to self-transition. Many systems, however, do not have distinct observation distributions associated with every action, like the locking mechanisms of Baum et al. (2017) or many standard POMDPs in the literature, like Tiger (Kaelbling et al., 1998).

We investigate the relationship between PSRs and tensor decomposition methods to learn a broader class of POMDPs than existing tensor methods. To connect the two approaches, we extend a result established by Carlyle & Paz (1971) and Balle et al. (2014) that states that PSRs learn transitions and observation matrices up to an unknown basis. We then reformulate tensor decomposition methods to estimate the unknown basis to recover the original basis. Our modification of tensor decomposition methods for hidden state inference allows us to simultaneously leverage all observation distributions from *all* actions with full-rank transition methods all at once, rather than a per-action basis like previous approaches (Azizzadenesheli et al., 2016; Guo et al., 2016). Should the collection of observation distributions of all full-rank actions be unique for each state, like Tiger, we may recover the full POMDP. Should there exist states that share the same set of observation distribution when aggregated across actions, we learn transitions between partitions of states, where states in a single partition share the same observation distributions over all actions.

Learning explicit transition and observation models is valuable because they enable model-based reasoning over environment dynamics. Whereas black-box PSRs only provide predictive likelihoods of observation sequences, access to explicit transition matrices and observation matrices allows for the specification of rewards after the model has been learned to direct planner behavior. Our experimental results suggest that our method can correctly learn partition-level transitions and observations and that these likelihoods are necessary to correctly direct agent behavior in POMDPs with very noisy observations.

## 2   PROBLEM SETTING

We assume that the ground truth system can be described as a discrete POMDP, represented by a 8-tuple $(\mathcal{S}, \mathcal{T}, \mathcal{A}, \mathcal{O}, \mathcal{Z}, b_0, R, \gamma)$. The set $\mathcal{S} = \{s^1, s^2, \dots\}$ is a discrete set of states, $\mathcal{A}$ is a discrete set of actions, and $\mathcal{O} = \{o^1, o^2, \dots\}$ is a discrete set of observations. $\mathcal{T} = \{T^a : a \in \mathcal{A}\}$ denotes a set of row-stochastic state transition matrices. The element $T^a_{ij} = \mathrm{P}(s_{t+1} = s^j | s_t = s^i, a_t = a)$ denotes the probability of transition to state $s^j$ from state $s^i$ after taking action $a$ at time $t$. The set $\mathcal{Z} = \{O^{ao} : (a, o) \in \mathcal{A} \times \mathcal{O}\}$ describes a collection of diagonal matrices, where $O^{ao}_{ii} = \mathrm{P}(o_t = o | s_t = s^i, a_t = a)$ denotes the emission probability of $o$ under action $a$ when leaving state $s^i$. The distribution $b_0 \in \Delta(\mathcal{S})$ describes the distribution over the initial state. The constant $\gamma \in (0, 1)$ is the reward discount factor.

The agent begins acting in a POMDP with access to the action and observation spaces $\mathcal{A}$ and $\mathcal{O}$. Under a uniform, memoryless random exploration policy $a_t \sim \mathrm{Unif}(\mathcal{A})$ for all $t \geq 1$, the agent collects a dataset $\mathcal{D}$, which is a long string of actions and observations $\mathcal{D} = (a_1, o_1, a_2, o_2, \dots)$. From this data, we wish the agent to estimate the number of hidden states $|\mathcal{S}|$, transition matrices $\hat{\mathcal{T}} = \{\hat{T}^a : a \in \mathcal{A}\}$, and observation matrices $\hat{\mathcal{Z}} = \{\hat{O}^{ao} : (a, o) \in \mathcal{A} \times \mathcal{O}\}$. We may also require the agent to learn a tabular reward $R$ function by including rewards as observations (Izadi & Precup, 2008). One way we evaluate the approach is by measuring the error of the estimated model parameters against those of the groundtruth POMDP. Another is by evaluating the performance agent behavior under a planning algorithm after the POMDP is inferred from $\mathcal{D}$. The last is by evaluating the behavior of a planner at a task designated by a user after the model has been learned.

## 3 LEARNING PREDICTIVE STATE REPRESENTATIONS

### 3.1 FORWARD, BACKWARD, AND HANKEL MATRICES

To estimate systems of hidden state, a natural place to start is to form an array that expresses the joint likelihoods between the observable random variables. A *Hankel matrix* is an instance of these arrays that encodes the joint likelihoods of past and future action-observation trajectories. In this section, we derive the Hankel matrix given knowledge of the ground truth POMDP. Our construction starts with two intermediate factors, called the *forward* and *backward* matrix, which we will multiply together to form the Hankel matrix.

The forward matrix **Forw** is a $\infty \times |\mathcal{S}|$ matrix whose rows are indexed by histories of action-observation sequences. The row indices are determined by choosing an ordering that enumerates all possible history sequences. A sensible ordering is to enumerate the action-observation sequences of length one first, then of length two, etc. In practice, while there are an infinite number of action-observation sequences, enumerating sequences up to a certain length is sufficient for computation. If $hist = (a_1, o_1, \ldots, a_t, o_t)$, then an entry $\mathbf{Forw}_{hist,i}$ expresses the *joint likelihood* of observing the history and landing in state $s^i$. The entire row of $hist$ may be computed by 'forward multiplying' the POMDP matrices that correspond to the action and observations in $hist$

$$\mathbf{Forw}_{hist,i} = \mathrm{P}(o_1, \ldots, o_t, s_{t+1} = s^i | a_1, \ldots, a_t), \tag{1}$$

$$\mathbf{Forw}_{hist,:} = b_0 \cdot O^{a_1 o_1} T^{a_1} \cdots O^{a_n o_n} T^{a_n}. \tag{2}$$

The backward matrix **Back** is a $|\mathcal{S}| \times \infty$ matrix whose columns are indexed by 'future' action-observation sequences. Following the PSR literature (Singh et al., 2004), we call these sequences *tests*. An ordering must be decided to determine the test indices in the same manner as the histories. Should $test = (a_{t+1}, o_{t+1}, \ldots, a_{t+n}, o_{t+n})$, then the entry $\mathbf{Back}_{i,test}$ expresses the likelihood of observing the test conditioned on beginning in state $s^i$. The entire column of $test$ can also be derived through forward multiplication of the POMDP matrices

$$\mathbf{Back}_{i,test} = \mathrm{P}(o_{t+1}, \ldots, o_{t+n} | a_{t+1} \ldots, a_{t+n}, s_{t+1} = s^i) \tag{3}$$

$$\mathbf{Back}_{:,test} = O^{a_{t+1} o_{t+1}} T^{a_{t+1}} \cdots O^{a_{t+n} o_{t+n}} T^{o_{t+n}} \cdot \mathbf{1} \tag{4}$$

where $\mathbf{1}$ is the vector where all entries are set to one.

The product of the forward and backward matrices results in the Hankel matrix, which we denote as $\mathcal{H}$. The matrix multiplication unconditions and then marginalizes out the intermediate hidden state. Given a history $hist = (a_1, o_1, \ldots, a_k, o_k)$ and a test $test = (a_{k+1}, o_{k+1}, \ldots, a_n, o_n)$, a Hankel matrix entry is the corresponding joint likelihood of receiving a full string of observations conditioned on taking a full string of actions, e.g.

$$\mathcal{H}_{hist,test} = \mathrm{P}(o_1, \ldots, o_n | a_1, \ldots a_n). \tag{5}$$

The Hankel matrix does not refer to the underlying hidden state of the POMDP and can be estimated from action-observation trajectories. If we had a long string of actions and observations $\mathcal{D}_n = (a_1, o_1, \ldots, a_n, o_n)$, the matrix $\mathcal{H}$ could be estimated by the *suffix-history approach*, taking frequency counts of subsequences of increasing lengths (Wolfe et al., 2005; Boots et al., 2011):

$$\hat{\mathcal{H}}_{hist,test} = \frac{\sum_{i=1}^{n-L} \mathbb{I}_{(a_i, o_i, \ldots, a_{i+L}, o_{i+L}) = hist \oplus test}}{\sum_{i=1}^{n-L} \mathbb{I}_{(a_i, \ldots, a_{i+L},) = \mathrm{acts}(hist \oplus test)}} \tag{6}$$

where $\mathrm{acts}(hist \oplus test)$ is the action sequence associated with $hist \oplus \mathcal{D}_{\mathrm{test}}$ and $L = |hist \oplus test| < n$.

This factorized construction is inspired by the construction of a related matrix, called the *System Dynamics Matrix*, by Singh et al. (2004).[1] It is important to note that expressing the Hankel matrix as a factorization of **Forw** and **Back** represents the system under a memoryless policy where future actions are independent of previous observations. To correctly estimate the matrix via Eq. 6, the data must also be collected under a memoryless policy, such as the uniform exploration policy as introduced in Sec. 2 (Bowling et al., 2006).

---

[1] The entries of the System Dynamics Matrix (SDM) are the likelihood of a given test *conditioned* on a history. Each row of the Hankel matrix is the same as the SDM except scaled by a constant (the likelihood of the history that indexes the row; Bacon et al. (2015)).

## 3.2 Transforming Predictive State Representations

Suppose we have a Hankel matrix $\mathcal{H}$, estimated in the limit of infinite data. Since the Hankel matrix $\mathcal{H}$ can be computed by multiplying two low-rank factors together, a natural first step of our method (and learning a PSR) is to compute a rank factorization of $\mathcal{H}$ (Boots et al., 2011; Balle et al., 2014). One way to achieve this factorization is to compute a singular-value decomposition of the Hankel matrix $\mathcal{H} = U\Sigma V^T$, where singular values under a specified threshold (and their corresponding orthogonal vector components) are dropped. The truncated SVD is converted into a *rank factorization* by computing $A = U\Sigma$ to be the left factor and $V^T$ to be the right factor. Crucially, since $A \cdot V^T$ and $\mathbf{Forw} \cdot \mathbf{Back}$ both form rank factorizations of $\mathcal{H}$ (according to assumptions in Sec 3.3), there must exist some invertible transformation $P$ such that $A = \mathbf{Forw} \cdot P$ and $P^{-1} \cdot \mathbf{Back} = V^T$ (see Appendix A.2).

Moving one step earlier in the Hankel construction (Sec. 3.1), we can relate transitions, observations, and initial distributions with the rank factors and the Hankel matrix using Eqs. 1-4. Let $hists^{ao}$ denote an ordered set of all history indices that end in action-observation pair $ao$, and $hists^{-ao}$ denote the same set with the same ordering but without the ending pair $ao$. From Eqs. 2 and 4, we observe for each $a \in \mathcal{A}, o \in \mathcal{O}$:

$$\mathcal{H}_{hists^{ao},:} = \mathbf{Forw}_{hists^{-ao},:} \cdot O^{ao}T^a \cdot \mathbf{Back} = A_{hists^{-ao},:} \cdot P^{-1}O^{ao}T^aP \cdot V^T \tag{7}$$

$$\mathcal{H}_{\varepsilon,:} = b_0^T \cdot \mathbf{Back} = b_0^T P \cdot V^T \tag{8}$$

$$\mathcal{H}_{:,\varepsilon} = \mathbf{Forw} \cdot \mathbf{1} = A \cdot P\mathbf{1} \tag{9}$$

After applying the Moore-Penrose inverse of $A$ and $V^T$ to solve Eqs. 7-9, we obtain the observation-transition product, initial belief, and final summation vector up to a similarity transform. The transformed initial belief $m_0 = b_0^T P$ is called the *initial vector* and the transformed summation vector $m_\infty = P\mathbf{1}$ is called the *final vector*. The product $M^{ao} = P^{-1}O^{ao}T^aP$ is called a linear PSR update matrix. Together, this collection of matrices and vectors forms a *linear PSR model* (Littman & Sutton, 2001; Boots et al., 2011). A PSR can be used to compute the likelihood of observations $o_1, o_2, \ldots, o_n$ under actions $a_1, \ldots, a_n$ by computing the product $\mathrm{P}(o_1, \ldots, o_n | a_1, \ldots, a_n) = m_0^T M^{a_1 o_1} \cdots M^{a_n o_n} m_\infty = b_0 P P^{-1} O^{a_1 o_1} T^{a_1} \cdot O^{a_n o_n} T^{a_n} P^{-1} P\mathbf{1}$, with appropriate normalizations for conditional calculations. With a few more details, the argument sketched above is a proof of a result of Carlyle & Paz (1971) and later Balle et al. (2014). The original result given by the authors was for probabilistic automata.

**Proposition 1.** *[Carlyle & Paz (1971); Balle et al. (2014)] Let $\mathcal{H} = AV^T$ be a rank factorization of a Hankel matrix $\mathcal{H}$ with $\mathrm{rank}(\mathcal{H}) = r$ formed from a POMDP with initial state $b_\pi$, transition matrices $\{T^a\}$ and observation matrices $\{O^{ao}\}$. Suppose $m_0, \{M^{ao}, \forall(a,o) \in \mathcal{A} \times \mathcal{O}\}, m_\infty$ are computed as in Eqs. 7, 8, and 9. Then there exists a nonsingular matrix $P \in \mathbb{R}^{r \times r}$ such that $P^{-1}M^{ao}P = O^{ao}T^a$ for all $a \in \mathcal{A}, o \in \mathcal{O}$, $m_0^T P = b_\pi$, and $P^{-1}m_\infty = \mathbf{1}$.*

## 3.3 Assumptions

Before we discuss how we estimate the similarity transform $P$, we must introduce a few important assumptions. First, we assume that under a memoryless random exploration-policy $a \sim \pi_{\exp}(\mathcal{A}), \pi \in \Delta(\mathcal{A})^2$ (which, in this paper, we take to be uniform), the induced Markov chain $(s_t, a_t, o_t)_{t \geq 0}$ is ergodic. The visitation distribution over states will converge to a stationary distribution $b_\pi$, which has nonzero support over the entire state space. Second, we also assume that $\mathbf{Forw}$, when limited to indices corresponding to one fewer than the maximum sequence length, has the same rank as the number of states (e.g. is full-rank), and that $\mathbf{Back}$ is also full-rank. This assumption is required to exclude POMDPs that have been shown to be computationally intractable to learn (Jin et al., 2020; Liu et al., 2022). For further discussion on the realism of systems learnable by our method, see Sec. 4.1.1.

Our assumptions have a few consequences on the estimated Hankel matrix. We only sample the starting state distribution $b_0$ at the start of the problem, so $b_0$ has little influence over the Hankel matrix. Instead, the Hankel matrix will take on the stationary distribution $b_\pi$ as the initial distribution instead. Furthermore, the rank of the resulting Hankel matrix will be *equivalent* to the number of states of the POMDPs in the restricted class that adhere to our assumptions, as opposed to the lower bound as is the case for general POMDPs. We formalize these consequences in Appendix A.

---

$^2\Delta(\mathcal{A})$ denotes the set of distributions over the discrete set $\mathcal{A}$.

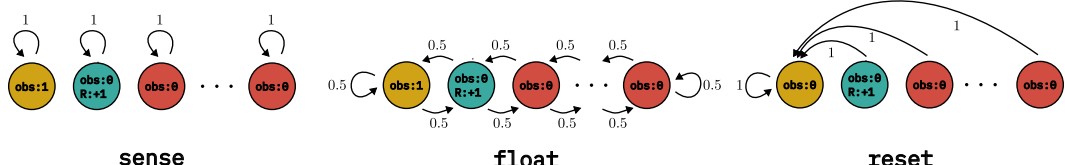

Figure 1: Sense-Float-Reset. Edges are labeled with transition probabilities, and nodes are labeled with observations and reward received upon *leaving* the state. The reward of a state is zero unless specified otherwise. Observability partitions are represented by node shade.

# 4 COMPUTING THE SIMILARITY TRANSFORM

In many problem scenarios, $P$ is nontrivial (e.g., not the identity matrix). The rank factors computed via SVD are orthogonal matrices, which is not always the case for **Forw** and **Back**. Furthermore, since rank factors may take real number values, $m_0$ and $\{M^{ao}\}$ are real-valued as well, and we cannot interpret their entries as state, transition, or observation emission likelihoods. These likelihoods are essential for downstream operations that direct agent behavior.

The key, then, is to recover the similarity transformation $P$. Once we know $P$, we can recover the original POMDP parameters by inverting the transform as expressed in Prop. 1. The observation and transition matrices can be recovered from products $O^{ao}T^a$ by computing the sums of the rows to form the diagonal of $O^{ao}$ and then normalizing the rows to form $T^a$. Our approach can recover $P$ up to a certain partition of states, which we introduce with an example.

**A running example.** Consider the POMDP illustrated in Figure 1, which is modified from the Float-Reset domain introduced by Littman & Sutton (2001). Like the original, the `float` action transitions the state up and down a line graph and will always emits an observation of `0`. The `reset` action, also identical to the original, deterministically sets the state to the left end of the graph. This action emits an observation of `1` if the state is already in the leftmost state and `0` otherwise. The observations of the `sense` action are the same as `reset`, except each state of the system does not change. We also augment the system a reward function; the agent obtains +1 reward for executing any action in the state adjacent to the reset state and zero reward otherwise. This system is challenging to learn due to its nontrivial partial observability. Aside from the two leftmost states (when treating rewards as observations), all other states in this POMDP have the same observation distributions, regardless of the action. Furthermore, the transition matrix corresponding to the `reset` action is singular since it is zero everywhere except for a single column of ones.

We wish to capture the difficulties of Sense-Float-Reset to discuss the main output of our algorithm in general terms. For arbitrary POMDPs, we group states that have the same observation distribution to form a *partition* of states. We call this grouping an *observability partition*. Of particular importance is the collection of observation distributions that correspond to actions with full-rank transition matrices. For the purpose of abbreviation, actions associated with full-rank transitions will be called *full-rank actions*, and we denote the entire set of full-rank actions as $\mathcal{A}_{full} \subseteq \mathcal{A}$. We call this alternate grouping restricted to action in $\mathcal{A}_{full}$ a *full-rank observability partition*.

## 4.1 RECOVERY UP TO A FULL-RANK OBSERVABILITY PARTITION

Our algorithm can estimate the similarity transform $P$ up to the *full-rank observability partition*, which we formalize in Theorem 1. Our statement is given in the regime of infinite data; for parameters introduced for finite data, see Appendix B.1.

**Theorem 1.** *Let $\mathcal{H}$ be a Hankel matrix of POMDP $(\mathcal{S}, \mathcal{T}, \mathcal{A}, \mathcal{O}, \mathcal{Z}, b_\pi, R, \gamma)$ that adheres to the assumptions in Sec. 3.3, where $b_\pi$ is the stationary distribution under a uniform random policy $a \sim \mathrm{Unif}(\mathcal{A})$. Let $S_\Pi \subset 2^{\mathcal{S}}$ be the full-rank observability partition of the POMDP. Let $A$ and $V^T$ be a rank factorization of $\mathcal{H}$, and $m_0$, $\{M^{ao} : a \in \mathcal{A}, o \in \mathcal{O}\}$, and $m_\infty$ be the linear PSR model as computed via Eqs. 7-9. Then there exists an algorithm on inputs $A$, $V^T$, $m_0$, $\{M^{ao}\}$, and $m_\infty$ that*

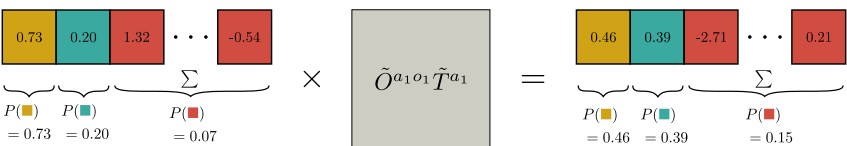

Figure 2: An illustration of Theorem 1 applied to Sense-Float-Reset. Summing indices over partitions, represented by box shades, will compute the likelihood of the system state in that partition.

*computes a nonsingular matrix $\tilde{P}$, such that if we let*

$$\tilde{b}_\pi^T = m_0^T \tilde{P} = b_\pi P^{-1} \tilde{P} \tag{10}$$

$$\tilde{O}^{ao} \tilde{T}^a = \tilde{P}^{-1} M^{ao} \tilde{P} = \tilde{P}^{-1} P O^{ao} T^a P^{-1} \tilde{P} \tag{11}$$

$$\tilde{b}_\infty = \tilde{P}^{-1} m_\infty = \tilde{P}^{-1} P \mathbf{1} \tag{12}$$

*then*

$$\sum_{s^i \in S} \tilde{b}_{\pi i} = \sum_{s^i \in S} b_{\pi i} \tag{13}$$

$$\sum_{s^i \in S} (\tilde{b}_\pi^T \tilde{O}^{a_1,o_1} \tilde{T}^{a_1} \dots \tilde{O}^{a_n,o_n} \tilde{T}^{a_n})_i = \sum_{s^i \in S} (b_\pi^T O^{a_1 o_1} T^{a_1} \dots O^{a_n o_n} T^{a_n})_i \tag{14}$$

$$\tilde{b}_\infty = \mathbf{1} \tag{15}$$

*for all $a_1, \dots, a_n \in \mathcal{A}$, $o_1, \dots, o_n \in \mathcal{O}$, and integer $n > 0$ and every partition set $S \in S_\Pi$.*

What Theorem 1 states is that we must sum over indices of the initial 'belief vector' to compute the likelihood the system is in a particular partition (Eq. 10). The same remains true when computing joint likelihoods between observations and the current state partition (Eq. 14; see Fig. 2 for a worked example for Sense-Float-Reset.). For POMDPs that have unique observation distributions across all actions, each state is in its own singleton partition, and we can recover the full similarity transform. Otherwise, we recover $P$ up to the full-rank observability partition. We note it is possible for us to recover some POMDPs that have fewer observations than states, since the collection of *distributions* over emitted observations across all actions must be distinct (see Appendix C.5.3 for examples).

### 4.1.1 ON THE RESTRICTIVENESS OF LEARNABLE SYSTEMS

To benefit from the result of Theorem 1, the systems to be learned must satisfy the assumptions stated in Sec. 3.3 and contain full-rank actions. Here, we discuss when these assumptions are satisfied.

**Full-Rank Transitions.** In automated manipulation, robot actions have a desired transition state but may also *fail* (a gripper misses a grasp, slips of a drawer handle, etc.). One way these actions have been modeled in robot planning systems is to designate a successful 'desired state' with some success likelihood $p_{succ}$, and have the system state 'fail' with some likelihood (causing a self-transition) (Kaelbling & Lozano-Pérez, 2013; Garrett et al., 2020). In POMDP terms, these types of actions can be simply modeled as the convex combination $p_{succ} T + (1 - p_{succ}) I$, where $T$ is a matrix with rows containing all zeros except for a single entry of 1 (the desired states), the identity $I$ indicates self-loop failure dynamics, and $p_{succ}$ the likelihood of an action succeeding. Under mild assumptions (e.g. $p_{succ} \neq 1/2, 1$), these actions are full-rank (see Appendix A.6).

**Ergodic Systems.** Since we stipulate that POMDPs must be learned from a single trajectory, it is reasonable that the robot must be able to explore every state to correctly learn transition dynamics and observation emissions. One condition of ergodicity, *irreducibility*, ensures that the system does not get trapped in a subset of states. Furthermore, in many robot manipulation scenarios, robots are given a passive 'sensing' action that only obtains an observation sample without causing a change to the system state (Kaelbling & Lozano-Pérez, 2013). The presence of these actions break any periodic cycles, the other condition of ergodicity.

## 4.2 Recovering Observation Distributions from Full-Rank Actions

We now introduce an algorithm that computes the similarity transform $\tilde{P}$. Our approach is a re-formulation of the tensor decomposition method (Anandkumar et al., 2012; Azizzadenesheli et al., 2016) for linear PSR models.

Our procedure begins by marginalizing out the observations in matrices $M^{ao}$, yielding the transitions $T^a$ up to similar transform $P$. This marginalization can be done by summing all matrices $M^{ao}$ over all $o \in \mathcal{O}$ for some fixed $a \in \mathcal{A}$:

$$\sum_{o \in \mathcal{O}} M^{ao} = P \bigg( \sum_{o \in \mathcal{O}} O^{ao} T^a \bigg) P^{-1} = PT^a P^{-1} \tag{16}$$

With a slight abuse of notation, we denote $P^{-1} T^a P$ as the matrix $M^a$. The next step of our procedure continues with transitions that are full-rank, which can easily be determined by a threshold test on the singular value decomposition on all matrices $M^a$. Let $\mathcal{M}_{full} = \{ P^{-1} T^a P : a \in \mathcal{A}_{full} \}$ be the set of full-rank transitions. Next, we compute the observation matrices associated with the full-rank actions. For each $M^a \in M_{full}$ and $o \in \mathcal{O}$ we compute

$$M^{ao} \cdot M^{a-1} = PO^{ao} T^a P^{-1} (PT^a P^{-1})^{-1} = PO^{ao} P^{-1}. \tag{17}$$

Since we know that all matrices $O^{ao}$ are diagonal, the eigenvalues of the matrices $M^{ao} M^{a-1}$ will be the diagonal entries of $O^{ao}$. If the entries of a particular $O^{ao}$ are unique, then the eigenvectors computed from an eigendecomposition of $M^{ao} M^{a-1}$ will recover the columns of $P$ up to a scalar factor. However, it is common to have repeated observation likelihoods across states for a single action (like all of Sense-Float-Reset), and an eigendecomposition may produce *any* spanning set of the invariant space corresponding to the repeated eigenvalue.

To reduce ambiguity, we wish to compute a *joint diagonalization* of all matrices $M^{ao} M^{a-1}$, which attempts to diagonalize each matrix with the same similarity transform. We apply a method of He et al. (2024). Their method exploits the fact that *sums* of matrices $\{ M^{ao} M^{a-1} : a \in \mathcal{A}_{full}, o \in \mathcal{O} \}$ do not change the invariant spaces spanned by eigenvectors of each matrix $M^{ao} M^{a-1}$. Suppose $\{ w^{ao} : a \in \mathcal{A}_{full}, o \in \mathcal{O} \}$ is a set of weights, then the weighted sum

$$\sum_{a \in \mathcal{A}_{full}, o \in \mathcal{O}} w^{ao} M^{ao} M^{a-1} = P \bigg( \sum_{a \in \mathcal{A}_{full}, o \in \mathcal{O}} w^{ao} O^{ao} \bigg) P^{-1} \tag{18}$$

is still diagonalizable by $P$. Should we choose *random* weights $w_{ao}$, then the eigenvalues will be distinct up to states that share the same observation distribution almost surely. He et al. (2024) recommends sampling these weights from the unit sphere $\mathbb{S}^{|\mathcal{A}_{full}| \cdot |\mathcal{O}| - 1}$.

**Lemma 1.** *Let weights $\{ w_{ao} : a \in \mathcal{A}_{full}, o \in \mathcal{O} \}$ be sampled i.i.d. with respect to* $\mathrm{Unif}(\mathbb{S}^{|\mathcal{A}_{full}| \cdot |\mathcal{O}| - 1})$ *and* $\Lambda = \sum_{a \in \mathcal{A}_{full}, o \in \mathcal{O}} w^{ao} O^{ao}$. *Then* $\Lambda_{ii} = \Lambda_{jj}$ *with prob. 1 if and only if* $O_{ii}^{ao} = O_{jj}^{ao}$ *for all $o \in \mathcal{O}$ and all $a \in \mathcal{A}_{full}$.*

When multiple states have the same observation distribution for all actions, the eigenvalues corresponding to those states will be the same, so their eigenvectors cannot be uniquely determined. Thus, the similarity transform $P'$ is nonunique when we have a nontrivial full-rank observability partition, the consequences of which we discuss in the next session.

## 4.3 Recovering Partition-Level Belief State Likelihoods and Transitions

The recovered similarity transform $P'$ formed by the eigenvectors of the random sum in Equation 18, but not the partition-level transitions. When the full-rank observability partition is nontrivial, the matrix $Q = P^{-1} P'$ is block-diagonal, with invertible blocks that correspond to states within the same partition (see Appendix A.4 for a proof). This matrix $Q$ prevents us from using $P'$ as the transform promised in Theorem 1. For example, when applying $P'$ as a similarity transform to the PSR vector $m_0$, a restriction to the subindices of the partition $S_1$ yields $[m_0 P']_{S_1} = [b_0^T P^{-1} P']_{S_1} = [b_0^T]_{S_1} Q_1$, so the sum of the entries is not a proper likelihood, violating Eq. 13 of Theorem 1.

To recover partition-level likelihoods and transitions, we look to the final vector of the linear PSR after applying the transform $P'$, e.g. $Pm_0 = P'^{-1} P\mathbf{1}$. Intuitively, by applying $\mathrm{diag}(Pm_0)$ as a similarity transform, we transform the final vector back to $\mathbf{1}$, recapturing a marginalization of the latent

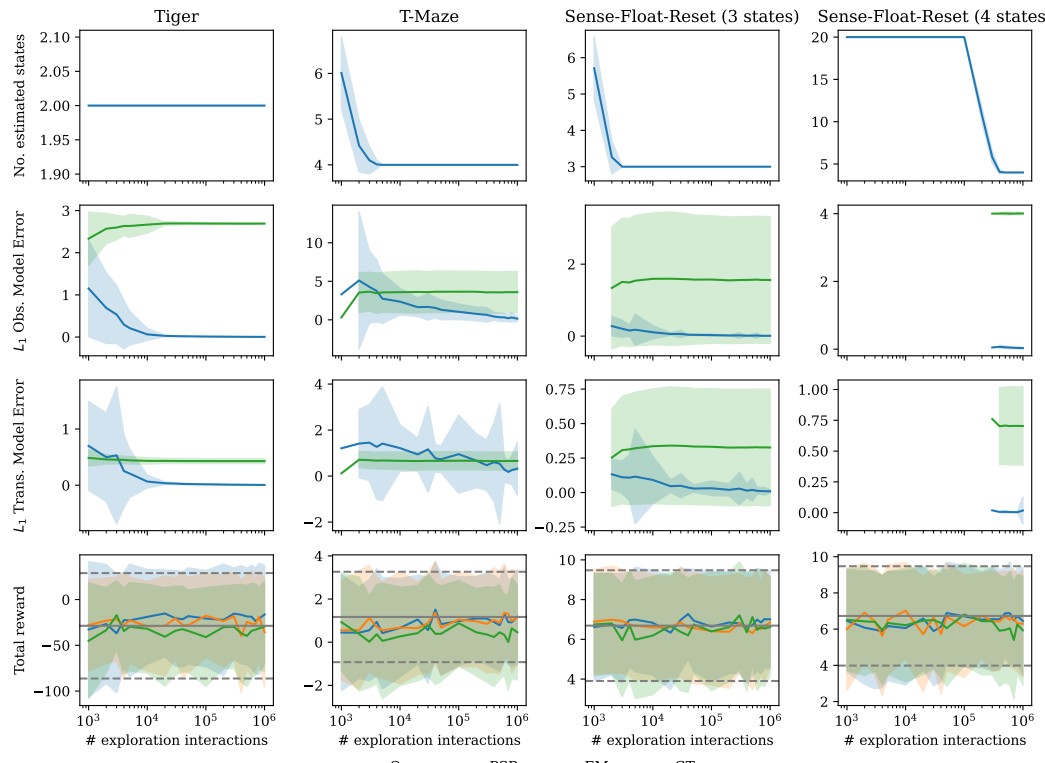

Figure 3: Error bars represent standard deviation over 100 seeds. The y-axis is scaled to make convergence visible. **Row 1:** Estimated number of states. **Row 2:** Obs. matrix error relative to ground truth. **Row 3:** Trans. matrix error. This error is only measurable once the estimated number of states matches that of ground truth, which truncates the curves. **Row 4:** Total reward from planner under different sampling strategies (see Appendix C.3).

state variable. To avoid scenarios where $P'^{-1}m_0$ has entries of zero, we perform a pre-processing step by multiplying the system with a random block-diagonal rotation matrix $R$, whose blocks correspond to the full-rank observability partition. We take the transform $\mathrm{diag}(RP'^{-1}m_\infty)RP'^{-1}$ as the similarity transform $\tilde{P}$ that satisfies Theorem 1 (see Appendix A.5 for proof of correctness).

## 5 EXPERIMENTS

Our experiments evaluate the fidelity of the learned POMDP models and explore the advantages of estimating transition and observation likelihoods. We seek to know, empirically, how quickly the learned observation matrices (and fine-grained transition matrices, if available) converge to ground truth values. We also wish to know whether the performance of the planning model is impaired by errors in the estimated similarity transform. Lastly, we evaluate whether the transition and observation likelihood estimates can be leveraged to specify a reward function to elicit desired behavior from a planner. All experiments are compared against linear PSRs and an Expectation-Maximization (EM) baseline (Rabiner, 1989; Shatkay & Kaelbling, 2002) with a number of states determined by the number of components of the truncated SVD when learning a linear PSR.

For our planning experiments, we verify our approach on several standard POMDPs: Tiger (Kaelbling et al., 1998), T-Maze (with a truncated corridor) (Bakker, 2001), and Sense-Float-Reset. To allow the agent to collect an arbitrary-length string of data in all domains, we modify T-Maze to choose the next state randomly from the initial state distribution instead of terminating the sequence of interactions. Appendices B.1 and C contain details on the parameters of the learning algorithm and planner. Rewards of the original POMDPs have been learned as observations for planning.

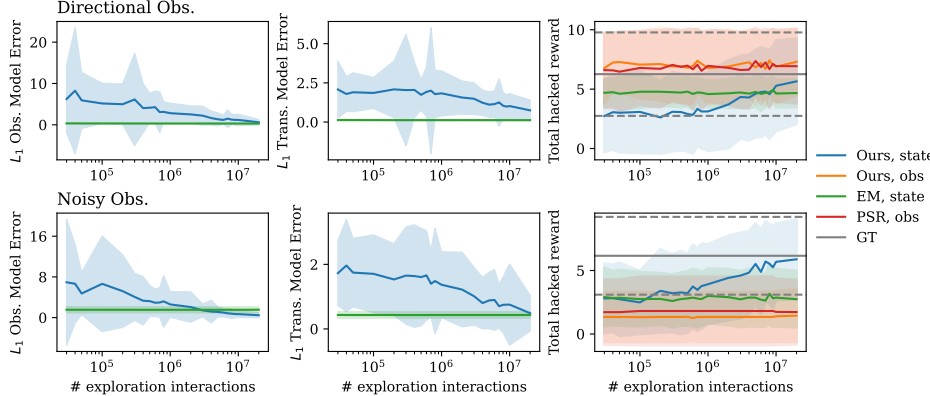

Figure 4: The agent receives +1 reward for each timestep in the designated goal state (middle of hallway). 'Obs' refers to assigning rewards to action-observation pairs, whereas 'state' refers to assigning rewards to states. Error bars report standard deviation over 100 seeds.

For our reward-specification experiments, we introduce two novel domains (*noisy hallway* and *directional hallway*) whose observation and transition matrices can be fully recovered by our method. The domains share transition dynamics on a three-state 'hallway,' in which the actions include noisily translating `left` and `right`, choosing to deterministically `stay` in the current cell, or performing a `reset` to a uniform random state. The observation space is also shared; the agent may noisily observe whether it is on the `left-end` or `right-end` of the hallway. The domains differ in the observation distributions of the middle states. In *noisy hallway*, the agent noisily observes the end of the hallway in the direction of commanded movement ('directional' observations). In *directional hallway*, the agent observes `left-end` or `right-end` with probability $1/2$ ('noisy' observations). For more details, see Appendix C.5.3.

**Convergence to true POMDP parameters.** In Figure 3, our results suggest that our method successfully recovers the underlying observation models through the $L_1$ error of learned observation and partition-level transition likelihoods against ground truth. EM consistently converges to a local minimum and does not obtain correct observation or transition likelihoods.

**Planning performance with the learned model.** To evaluate the performance of the planning model, we apply a standard sampling-based POMDP solver to the original ground truth POMDPs, learned PSRs, and learned POMDPs and compare the average yielded rewards. We use the sampling-based planning approach PO-UCT of Silver & Veness (2010) with the correction described by Shah et al. (2022). Ideally, planning performance should be the same across ground truth models, PSRs, and the learned partition-level POMDPs (see Appendix C.3 for discussion on rollout strategies for each model). Performance as a function of the number of action-observations collected is shown in Fig. 3, which we find to be similar across all models learned.

**Planning performance on specified rewards.** We explore whether the likelihoods and observations yielded by our algorithm can be leveraged to direct agent behavior after the model is learned. One case, motivated by automated planning in robotics, is to direct the agent to drive a system into a set of states as determined by the states' emitted observations by specifying a reward function (Boots et al., 2011). After learning a POMDP, we can analyze the learned observation matrices to find the states to emit positive reward. In the past, if a PSR did not learn a reward model, then rewards were determined solely by observations (Boots et al., 2011). Otherwise, the entire model must be relearned to estimate a reward model that depends on state (Izadi & Precup, 2008).

Our evaluations of this experiment are carried out on the two noisy hallway domains, where we attempt to direct the agent to drive the POMDP to the 'middle' hallway state with ambiguous observations. We compare the strategies of assigning rewards to observations and assigning rewards to states. In the directional domain, we assign +1 reward to action-observation pairs (`left`,`end-left`) and (`right`,`end-right`) for the former strategy, and assign +1 reward to the state whose maximum likelihood observations under `left` and `right` are their corresponding hallway ends for the latter. For the noisy environment, we reward the same action-observation

pairs as the directional environment and also (`left`,`end-right`) and (`right`,`end-left`) for the former strategy, and add $+1$ reward to the state that maximizes the sum of entropy of observation distributions across all actions for the latter. The former strategy is evaluated on PSRs and POMDPs, whereas the latter is evaluated on learned POMDP models only. Performance is judged on the number of timesteps the agent spends in the desired states.

Results can be found in Figure 4. In the directional domain, models that use the first strategy allow the planner to drive the middle state because it is easily identified by the observations received under `left` and `right`. The second strategy performs poorly due to slow convergence of transition matrices (see Appendix C.4). In the noisy domain, the uniform belief state and belief state that places all mass on the middle of the hallway yield the same mixture observation distribution weighted by the belief, which does not elicit the correct behavior from the planner. The planner that uses the rewards emitted from the highest-entropy state performs well after the transition matrices begin to converge. This additional flexibility highlights that learning POMDPs maintains all the advantages of PSRs and obtains the flexibility to exploit observation and transition likelihood models.

## 6 RELATED WORK

Spectral methods are a common technique for learning partially-observable dynamics in the theory of RL literature (Liu et al., 2022; Zhan et al., 2022). These models are sufficient to serve as black-box models for model-based RL but do not afford likelihood estimates for other computations that require general inference operations related to the latent state. Spectral methods to model learning have been applied to related settings, including linear time-invariant system identification (Ho & Kalman, 1966; Oymak & Ozay, 2022). Other alternate approaches for learning POMDPs have also been explored. Toro Icarte et al. (2019) applies mixed-integer linear programming to learn an automaton to describe transition data. The learning problem has been framed as an automaton learning problem by Angluin (1987) to accept a particular *language* with strings given as data (Brafman & De Giacomo, 2019; Ronca et al., 2022). Others have resorted to inductive logic schemes (Amir & Chang, 2008; Silver et al., 2021). These methods usually assume that transitions or observations made by the agent are deterministic. Other approaches have relaxed assumptions to stochastic observations but still assume deterministic transitions (Dean et al., 1995).

Recurrent deep-learning-based architectures (Wang et al., 2023; Allen et al., 2024), can learn to make future predictions of system behavior from histories of actions and observations. Recurrent neural nets perform particularly well, unlike Transformers, which represent a fixed circuit that cannot maintain memory internally (Lu et al., 2024). These recurrent models are use specialized training objectives that encourage networks to learn how to summarize histories observed by the agent (Agarwal et al., 2021; Allen et al., 2024) or have access to priveleged full-state observable information during training (Wang et al., 2023). Like PSRs, the representation of the hidden state learned by these models is , and cannot readily provide likelihood models for probabilistic inference.

## 7 CONCLUSION AND FUTURE WORK

We present a method that learns discrete POMDP parameters from an action-observation sequence gathered under a random exploration policy up to a partition of the state space. Our approach applies tensor decomposition methods to estimate a similarity transform to transform a PSR model to a basis where observation and transition likelihoods can be recovered. In domains where each state has a unique observation distribution aggregated across all full-rank actions, we recover the true POMDP. Otherwise, we learn the transitions between full-rank observability partitions of the state space.

In the future, we intend to improve our method to scale to larger problems, but in class and scale. Removing the restriction of learning observations from full-rank actions, or learning full transitions despite the presence of a nontrivial observability partition would be desirable. Another direction is to improve our approach to scale to larger POMDPs. Matrix-completion methods under low-rankness assumptions could help the algorithm infer *missing* entries in the Hankel matrix. Additional future work is to expand the theoretical foundations of our algorithm. Carefully studying our algorithm under a PAC-learning framework would contribute to our understanding of the computational complexity of learning POMDPs in general.

**Reproducibility Statement:** All omitted proofs to derive the theoretical claims in this paper have been included in Appendix A. Discussion regarding algorithm parameters selected (Hankel matrix size, rankness thresholds, UCT constants, etc.) and experimental domains can be found in Appendices B and C. Code, including environments and the learning algorithm, will be made available should our work be accepted for publication.

**Ethics Statement:** This paper discuses the derivation of a novel learning algorithm and application in toy experimental planning domains, so we do not believe there is any cause for ethical concern of our work.

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

## A  OMITTED PROOFS

### A.1  FORMALIZATION OF ASSUMPTIONS

Here, we formualize the consequences of the assumtions discussed in Section 3.3.

**Lemma 2.** *Let $(\mathcal{S}, \mathcal{T}, \mathcal{A}, \mathcal{O}, \mathcal{Z}, b_0, R, \gamma)$ be a POMDP. Let $\mathcal{D}_n = (a_1, o_1, \ldots, a_n, o_n)$ for $n > 0$, where $a_i \sim \mathrm{Unif}(\mathcal{A})$ for all $i$. Suppose the POMDP admits the assumptions outlined above. Furthermore, let $\hat{\mathcal{H}}$ be the 'empirical' Hankel matrix as computed in Eq. 6. Consider the Hankel matrix in the 'limit of infinite data,' where $\mathcal{H} = \lim_{n \to \infty} \hat{\mathcal{H}}$. Then $\mathcal{H}$ is the Hankel matrix of POMDP $(\mathcal{S}, \mathcal{T}, \mathcal{A}, \mathcal{O}, \mathcal{Z}, b_\pi, R, \gamma)$ and $\mathrm{rank}(\mathcal{H}) = |\mathcal{S}|$.*

For our proof, we rely on a fundamental result on the convergence to stationary distributions of ergodic Markov chains.

**Theorem 2** (Ergodic Theorem, Norris (1997))**.** *Let $T$ be ergodic with stationary distribution $\pi$, and let $b_0$ be any initial distribution. Let $(X_n)_{n \geq 0}$ be a Markov chain with respect to $T$ with initial distribution $b_0$. Then, for any bounded function $f : \mathcal{S} \to \mathbb{R}$ we have*

$$\mathrm{P}\left(\lim_{n \to \infty} \frac{1}{n} \sum_{k=1}^{n-1} f(X_k) = \bar{f}\right) = 1$$

*where*

$$\bar{f} = \sum_{i \in \mathcal{S}} \pi_i f_i,$$

*and $b_\pi$ is the stationary distribution of $T$.*

We also require knowledge of the stationary distribution of Markov chains created as a 'sliding window' of another ergodic Markov chain.

**Lemma 3.** *Let $(Y_t)_{t \geq 0}$ be a Markov chain with ergodic transition matrix $T$, with stationary distribution $\pi$, over state space $\mathcal{Y} = \{1, \ldots, k\}$. Then the Markov chain $(Y_t, Y_{t+1}, \ldots Y_{t+n-1})_{t \geq 0}$ is also ergodic, with stationary distribution $\tilde{\pi}(i_1, \ldots i_n) = \pi_{i_0} T_{i_0, i_1} \ldots T_{i_{n-2}, i_{n-1}}$.*

*Proof.* The transitions of this Markov chain can be expressed as

$$\mathrm{P}(i_{t+n}, \ldots, i_{t+1} | i_{t+(n-1)}, \ldots, i_t) = T_{i_{t+n-1}, i_{t+n}}$$

and zero if the indices do not follow the form above. We verify the stationary distribution claimed in the conclusion of the statement. Suppose that $(j_1, \ldots, j_n)$ is a state of the Markov chain $(Y_t, Y_{t+1}, Y_{t+n-1})$. When we apply the transition likelihoods above, we find that

$$\sum_{i_1, \ldots, i_n \in \mathcal{Y}} \tilde{\pi}(i_1, \ldots i_n) \, \mathrm{P}(Y_{t+1} = j_1, \ldots, Y_{t+n} = j_n | Y_t = i_1, \ldots, Y_{t+n-1} = i_n)$$

$$= \sum_{i_1} \pi(i_1) T_{i_1, j_1} \ldots, T_{j_{n-1}, j_n}$$

$$= \pi(j_1) T_{j_1, j_2} \ldots T_{j_{n-1}, j_n} = \tilde{\pi}(j_1, \ldots, j_n).$$

The second line applies the definition of a transition above and the proposed stationary distribution, and the third line uses the fact that $\pi$ is the stationary distribution of $T$. $\qquad \square$

We now have all the tools we need to prove Lemma 2.

*Proof.* **The Hankel matrix takes on the stationary distribution $b_\pi$ as the stationary distribution**:

We first consider the Markov chain $(X_t)_{t \geq 0}$, where $X_t = (s_t, a_t, o_t)$. Per our assumptions in Section 3.3, we assume that the Markov chain $(X_t)_{t \geq 0}$ over the state space $\mathcal{X} = \{(s^i, o^j, a^k) \in \mathcal{S} \times \mathcal{O} \times \mathcal{A} : \mathrm{P}(o^k | s^i a^k) > 0\}$ is ergodic. Suppose that its stationary distribution is $p_\pi$. We denote $p_\pi(s, a, o)$ to be the likelihood of the Markov chain $(s, a, o)$ under $p_\pi$. Of interest is the marginal stationary distribution of the *POMDP* state $s \in \mathcal{S}$, under $p_\pi$, which we denote as the vector $b_\pi$, where $(b_\pi)_i = \sum_{a \in \mathcal{A}, o \in \mathcal{O}} p_\pi(s, a, o)$.

We observe that for any $(a_0, o_0, s_0, \ldots, a_{k-1}, o_{k-1}, s_{k-1}) \in \mathcal{X}^k$, under Lemma 3 and Theorem 2,

$$
\lim_{n \to \infty} \frac{1}{n-k} \sum_{i=k}^{n} \mathbb{I}_{a_0, o_0, s_0, \ldots, a_{k-1}, o_{k-1}, s_{k-1} = X_{i-k}, \ldots, X_i}
$$
$$
= p_\pi(a_0, o_0, s_0) \prod_{i=0}^{k-1} \mathrm{P}(X_{i+1} = s_{i+1}, o_{i+1}, a_{i+1} | X_i = s_i, o_i, a_i) \tag{19}
$$

*almost surely* for any integer $k \geq 1$. Let P corresponds to the law induced by the stationary distribution $p_\pi$ and transition and observation models of the POMDP. We will now use Eq. 19 as a way to understand the convergent values of the Hankel matrix.

Let $\bar{\mathcal{D}}_n = (s_1, a_1, o_1, \ldots, s_n, a_n, o_n)$ be a sequence of the induced Markov chain, and let $\mathcal{D}_n = (a_1, o_1, \ldots, a_n, o_n)$ be the same dataset with the state variable omitted. Let $hist = (a^{j_1}, o^{k_1}, \ldots, a^{j_t}, o^{k_t})$ and $test = (a^{j_{t+1}}, o^{k_{t+1}}, \ldots, a^{j_L}, o^{k_L})$ be action-observation sequences, length $L < n$. Then, we may evaluate the empirical Hankel matrix $\hat{\mathcal{H}}$ using Eq. 19.

$$
\hat{\mathcal{H}}_{hist, test} = \frac{\sum_{i=1}^{n-L} \mathbb{I}_{(a_i, o_i, \ldots, a_{i+L}, o_{i+L}) = hist \oplus test}}{\sum_{i=1}^{n-L} \mathbb{I}_{(a_i, \ldots, a_{i+L},) = (a^{j_1}, \ldots, a^{j_L})}}
$$
$$
= \frac{1/(n-L)}{1/(n-L)}
$$
$$
\cdot \frac{\sum_{s^{m_1}, \ldots, s^{m_L}} \sum_{i=1}^{n-L} \mathbb{I}_{(a_i, o_i, s_i \ldots, a_{i+L}, o_{i+L}, s_{i+L}) = (a^{j_1}, o^{k_1}, s^{m_1}, \ldots, a^{j_L}, o^{k_L}, s^{m_L})}}{\sum_{o^{k_1} s^{k_1}, \ldots, o^{k_L} s^{k_L}} \sum_{i=1}^{n-L} \mathbb{I}_{(a_i, o_i, s_i, \ldots, a_{i+L}, o_{i+L}, s_{i+L}) = (a^{j_1}, o^{k_1}, s^{m_1}, \ldots, a^{j_L}, o^{k_L}, s^{m_L})}}
$$

Taking the limit of $n$ to infinity on both sides and applying Eq. 19 (noting the denominator is nonzero almost surely under a uniform random exploration policy) yields

$$
\lim_{n \to \infty} \hat{\mathcal{H}}_{hist, \mathcal{D}_{\text{test}}} = \mathrm{P}(o^{k_1}, \ldots, o^{k_L} | a^{j_1}, \ldots a^{j_L})
$$
$$
= \sum_{s^{m_1}, \ldots, s^{m_L}, s^{m_{L+1}}} \mathrm{P}(s^{m_1}, o^{k_1}, \ldots, s^{m_L}, o^{k_L}, s^{m_{L+1}} | a^{j_1}, \ldots a^{j_L})
$$
$$
= \sum_{s^{m_1}, \ldots, s^{m_L}, s^{m_{L+1}}} \mathrm{P}(s^{m_1}) \mathrm{P}(o^{k_1}, s^{m_2} | s^{m_1} a^{m_1}) \cdots \mathrm{P}(s^{m_{L+1}}, o^{k_L} | s^{m_L} a^{j_L})
$$
$$
= b_\pi O^{a^{j_1} o^{k_1}} T^{a^{j_1}} \cdots O^{a^{j_L} o^{k_L}} T^{a^{j_L}} \cdot \mathbf{1}
$$

The last line unpacks the probability law P introduced by Eq. 19 back into matrix notation. We can see that splitting the product fo the last line above over $hist$ and $test$ will reproduce individual rows and columns of **Forw** and **Back**, respectively. Finally, we observe that **Forw** has taken on the distribution $b_\pi$ as the initial vector in the product.

**The Hankel matrix is full-rank**: Since we know a submatrix of **Forw** formed a subselection of rows is full-rank then the full forward matrix **Forw** is full-rank as well. Thus, we know that both of the **Back** and **Forw** are full-rank. This means we can find $|S|$ linearly-independent rows of **Back** and $|S|$ linearly-independent columns of **Forw**, which we assemble into submatrices $K$ and

$W$ respectively. We observe, then, that $K$ and $W$ are full-rank and square. The product $K \cdot W$, then, must also be full-rank and square. When we multiply $\mathbf{Forw} \cdot \mathbf{Back} = \mathcal{H}$, we observe, then, that $K \cdot W$ is a submatrix of $\mathcal{H}$. Then we know that

$$|S| = \text{rank}(K \cdot W) \leq \text{rank}(\mathcal{H}) \leq \min(\text{rank}(\mathbf{Forw}), \text{rank}(\mathbf{Back})) = |S|,$$

so $\text{rank}(H) = |S|$. $\qquad\square$

### A.2 PROOF OF PROPOSITION 1

We have two rank factorizations of $\mathcal{H}$: $\mathbf{Forw} \cdot \mathbf{Back} = A \cdot V^T = \mathcal{H}$. Since all matrix factors involved are full-rank, we may take the Moore-Penrose inverse of $\mathbf{Forw}$ and $\mathbf{Back}$, which results in $\mathbf{Forw}^\dagger A \cdot V^T \mathbf{Back}^\dagger = I$. Then $\mathbf{Forw}^\dagger A$ is nonsingular and $V^T \mathbf{Back}^\dagger$ is its inverse.

We take the product $(\mathbf{Forw}^\dagger A)$ to be $P$. A consequence of the assumptions in Section 3.3 is that $A_{hists^{-ao},:}$ is full-rank for all $a \in \mathcal{A}$ and $o \in \mathcal{O}$. Thus, we could have repeated the argument above, but replacing $A$ with $A_{hists^{-ao}}$ and $\mathcal{H}$ with $\mathcal{H}_{hists^{-ao},:}$, and find that $P = \mathbf{Forw}^\dagger_{hists^{-ao},:} A_{hists^{-ao}}$.

What remains is to show that we can apply $P$ to recover the POMDP initial belief, observation matrices, transition matrices, and final summing vector from the linear PSR models. Let $a \in \mathcal{A}$ and $o \in \mathcal{O}$. Following Eq. 7, we know that $M^{ao} = A^\dagger_{hists^{-ao},:} \cdot \mathcal{H}_{hists^{ao},:} \cdot V^{T^\dagger}$. If we apply $P$ as a similarity transform, we find that

$$P^{-1}M^{ao}P = P^{-1}A^\dagger_{hists^{-ao},:} \cdot \mathcal{H}_{hists^{ao},:} \cdot V^{T^\dagger}P$$

$$= P^{-1}A^\dagger_{hists^{-ao},:} \cdot \mathbf{Forw}_{hists^{-ao},:} \cdot O^{ao}T^a \cdot \mathbf{Back} \cdot V^{T^\dagger}P$$

$$= P^{-1}P \cdot O^{ao}T^a P^{-1}P = O^{ao}T^a$$

The initial belief vector $b_0$ and all ones vector $\mathbf{1}$ can be recovered from the initial vectors $m_0$ and $m_\infty$ in the same manner (they only feature an inversion of $P$ on either the left or right sides, but not both as above). $\qquad\square$

### A.3 PROOF OF LEMMA 1

The 'only if' direction is immediate. We prove the 'if' direction by proving its contrapositive.

Fix $i, j$ such that $1 \leq i < j \leq |\mathcal{S}|$. Suppose that there exists an $a \in \mathcal{A}_{full}$, $o \in \mathcal{O}$ such that $O^{ao}_{ii} \neq O^{ao}_{jj}$. Let $(ao)_1, \ldots, (ao)_{|\mathcal{A}_{full}| \cdot |\mathcal{O}|}$ be an ordering on $\mathcal{A}_{full} \times \mathcal{O}$. Let $a, b$ be two $|\mathcal{A}_{full}| \cdot |\mathcal{O}|$-dimensional vectors such that $a_k = O^{(ao)_k}_{ii}$ and $b_k = O^{(ao)_k}_{jj}$ for all $1 \leq k \leq |\mathcal{A}_{full}| \cdot |\mathcal{O}|$. Then we know that $a \neq b$.

Consider the event that $w \in \mathbb{S}^{|\mathcal{A}_{full}| \cdot |\mathcal{O}|}$, such that $\Lambda_{ii} = \sum_k^{|\mathcal{A}_{full}| \cdot |\mathcal{O}|} w_k O^{(ao)_k}_i i$ and $\Lambda_{jj} = \sum_k^{|\mathcal{A}_{full}| \cdot |\mathcal{O}|} w_k O^{(ao)_k}_j j$ are equivalent. Written in terms of the notation introduced above, we have that $\langle a - b, w \rangle = 0$. This means that $w$ must be contained in the hyperplane $H = \{x \in \mathbb{R}^{|\mathcal{A}_{full}| \cdot |\mathcal{O}|} : \langle x, a - b \rangle = 0\}$, which also passes through the origin. We recognize that $H \cap \mathbb{S}^{|\mathcal{A}_{full}| \cdot |\mathcal{O}|}$ is a $|\mathcal{A}_{full}| \cdot |\mathcal{O}| - 2$-dimensional submanifold (a lower-dimensional sphere) of $\mathcal{S}^{|\mathcal{A}_{full}| \cdot |\mathcal{O}|}$. We know that the measure of this submanifold under the induced uniform measure on $\mathbb{S}^{|\mathcal{A}_{full}| \cdot |\mathcal{O}| - 1}$ from the Lebesgue measure on $\mathbb{R}^{|\mathcal{A}_{full}| \cdot |\mathcal{O}| - 1}$ is zero (Lee, 2012). Thus, the probability of sampling $w$ so that $\Lambda_{ii} = \Lambda_{jj}$ is zero as well. Therefore, the complement of this event, that $\Lambda_{ii} \neq \Lambda_{jj}$, must occur with probability one. $\qquad\square$

*Remark.* By a similar argument above, we obtain that $\text{diag}(\Lambda) \neq \mathbf{0}$ with probability 1, where $\mathbf{0}$ is the zero vector. The argument replaces the discussion of the vector $a - b$, as constructed above, with the individual vectors $a$ or $b$.

### A.4 PROOF THAT SIMILARITY TRANSFORM IS RECOVERED UP TO BLOCK-DIAGONAL MATRIX

First, we formalize the claim made in Sec. 4.3.

**Lemma 4.** *Let $P'$ be the similarity transform as determined by an eigendecomposition of the random sum of Eq. 18. Without loss of generality, permute the columns of $P'$ and $P$ so that states in the same full-rank observability partition are in consecutive indices. Then*

$$P^{-1}P' = \begin{pmatrix} Q_1 & 0 & \cdots & 0 \\ 0 & Q_2 & \cdots & 0 \\ 0 & 0 & \ddots & 0 \\ 0 & 0 & & Q_k \end{pmatrix} \tag{20}$$

*where the blocks $Q_i \in \mathbb{R}^{|S_i| \times |S_i|}$ are nonsingular w.p. 1, where $|S_i|$ is the $i^{th}$ partition in the permuted index ordering.*

Let $X$ denote the random sum as expressed in Eq. 18, and let $P\Lambda P^{-1}$, $P'\Lambda P'^{-1}$ be the two diagonalizations as discussed in Section 4.2. Let $S_\Pi = \{S_1, \ldots, S_k\}$ be the full-rank observability partition. By Lemma 1, then $\Lambda_{ii} = \Lambda_{jj}$ for all $s^i, s^j$ in the same partition as $S \in S_\Pi$. Furthermore, by the remark in Section A.3, we know that $\text{diag}(\Lambda) \neq \mathbf{0}$ with probability 1, where $\mathbf{0}$ is the zero vector.

Suppose the indices of these matrices are ordered as stated in the hypothesis. Since $X = P\Lambda P^{-1} = P'\Lambda P'^{-1}$, then we have that $P'^{-1}P\Lambda = \Lambda P'^{-1}P$ (e.g. $P'^{-1}P$ and $\Lambda$ commute). Examining the entries of equation $P'^{-1}P\Lambda - \Lambda P'^{-1}P=0$ yields that $(\Lambda_{ii} - \Lambda_{jj})(P'^{-1}P)_{ij} = 0$. If $s^i$ and $s^j$ are contained in separate partitions, then $\Lambda_{ii} - \Lambda_{jj} \neq 0$, so $P'^{-1}P_{ij} = 0$. Thus, $P'P^{-1}$ has the desired block-diagonal structure. Since we know that both $P^{-1}$ and $P'$ are invertible, so must be $P'^{-1}P$. Thus, we know the blocks are invertible as well. $\square$

### A.5 PROOF OF THEOREM 1

Section 4.3 claims that applying the matrix $P'R\,\text{diag}(R^TP'^{-1}m_\infty)$ is a similarity transformation $\tilde{P}$ that satisfies the implication of Theorem 1. As a reminder, $P'$ are the eigenvectors from the eigendecomposition of the matrix in Eq. 18 and $R$ is a random block-diagonal rotation matrix with the same block structure as $PP'^{-1} = Q$ (Lemma 4), whose blocks are distributed over the Haar measure over the corresponding copy of $SO(n)$.

We begin our argument by first applying the similarity transformation $\tilde{P}$ to a learned PSR $m_0$, $\{M^{ao} : a \in \mathcal{A}, o \in \mathcal{O}\}$ and $m_\infty$. We find that

$$m_0\tilde{P} = b_\pi QR\,\text{diag}(R^TQ^{-1}\mathbf{1}) \tag{21}$$

$$\tilde{P}^{-1}M^{ao}\tilde{P} = \text{diag}(R^TQ^{-1}\mathbf{1})^{-1}R^TQ^{-1} \cdot (T^aO^{ao}) \cdot QR\,\text{diag}(R^TQ^{-1}\mathbf{1}) \tag{22}$$

$$\tilde{P}^{-1}m_\infty = \text{diag}(R^TQ^{-1}\mathbf{1})^{-1}R^TQ^{-1} \cdot \mathbf{1} \tag{23}$$

If we unpack the block structure of $Q$, $R$, and $\text{diag}(R^TQ^{-1}m_\infty)$ in Equations 21 and 23, we find

$$[m_0\tilde{P}]_{S_i} = b_\pi Q_iR_i\,\text{diag}(R_i^TQ_i^{-1}[\mathbf{1}]_{S_i}) \tag{24}$$

$$[\tilde{P}^{-1}m_\infty]_{S_i} = \text{diag}(R_i^TQ_i^{-1}[1]_{S_i})^{-1}R_i^TQ_i^{-1} \cdot [\mathbf{1}]_{S_i} \tag{25}$$

where $R_i$ and $Q_i$ are the blocks associated with the full-rank observability partition $S_i$.

First, we must justify that the relations expressed in Equations 21-25 are well-defined. We already know $R$ and $Q$ are nonsingular. We must then show all entries of the vector $R^TQ^{-1}\mathbf{1}$ are nonzero to allow for the existence of $\text{diag}(R^TQ^{-1}\mathbf{1})$. This fact is a consequence of known properties of the Haar measure over special orthogonal matrices (Meckes, 2019, Section 1.2). Fix a full-rank observability partition $S_i$. It is known that corresponding rotation matrix blocks $R_i^T$ and $R_i$ are identically distributed with respect to the Haar measure on $SO(|S_i|)$ (Meckes, 2019, pg. 18). Furthermore, since $Q_i$ is nonsingular, we know that $Q_i^{-1}\mathbf{1}$ is not the zero vector. Thus, it is also known that the random vector $R_i^T \cdot (Q_i^{-1}[\mathbf{1}]_{S_i})$ is uniformly distributed over the $(|S_i| - 1)$-sphere with radius $\left\| Q_i^{-1}[\mathbf{1}]_{S_i} \right\|_2$ (Meckes, 2019, pg. 19-20, 26). By the same argument discussed in the proof of Lemma 1, the entries of $R_i^TQ_i^{-1}[\mathbf{1}]_{S_i}$ must be nonzero with probability one. By taking a union bound over all full-rank observability partitions, *all* entries of $R^TQ^{-1}\mathbf{1}$ must be nonzero with probability one as well.

What remains is to prove the correctness of the relations 13-15 in Theorem 1. The expression that we obtain $\tilde{P}^{-1}m_\infty = \mathbf{1}$ is immediate from Equations 23 and 25. Next, justify Equation 13. Fix an full-rank observability partition $S_i$. Then

$$\sum_{i \in S_i} [\tilde{b}_\pi]_i = [m_0 \tilde{P}]_{S_i}^T \cdot [\mathbf{1}]_{S_i}$$

$$= [b_\pi^T]_{S_i} Q_i R_i \operatorname{diag}(R_i^T Q_i^{-1}[\mathbf{1}]_{S_i}) \cdot \operatorname{diag}(R_i^T Q_i^{-1}[1]_{S_i})^{-1} R_i^T Q_i^{-1} \cdot [\mathbf{1}]_{S_i}$$

$$= [b_\pi^T]_{S_i} \cdot [\mathbf{1}]_{S_i}$$

$$= \sum_{i \in S_i} [b_\pi]_i$$

The second line applies Equations 24 and 25. The proof for Equation 14 is nearly the same, and can be reached by deriving an analagous expression to Equation 24 by first multiplying out the corresponding sequence of matrices $\tilde{P}^{-1}M^{ao}P$ and unpacking the block structure of $Q$ and $R$ again. $\qquad\square$

### A.6 PROOF OF FULL-RANK TRANSITION CLAIM

We formally state the claim made in the deliberation of Section 4.1

**Proposition 2.** *Let $T$ be an $n \times n$ matrix, with rows that are all zeros except for a single entry of 1 per row. Let $p \in [0, 1)$, and $p \neq 1/2$. Then the convex combination $pT + (1 - p)I$ is nonsingular.*

*Proof.* We first observe that the proof is immediate if $p = 0$, so we focus on the case for $p \in (0, 1)$. Suppose, for the sake of contradition, that there exists $v \in \mathbb{R}^n$ such that matrix-vector product $(pT + (1 - p)I)\,v = 0$. This must be true if and only if

$$Tv = \left(\frac{p - 1}{p}\right)v,$$

or that $v$ is an eigenvector of $T$ with eigenvalue $(p - 1)/p$.

We claim that the eigenvalues of $T$ are either zero or roots of unity. If this claim is true, we arrive at a contradiction, because if $p \neq 1/2$ and $p \in (0, 1)$, then $(p - 1)/p$ cannot be equal $-1$.

We prove this claim by induction on the number of rows and columns. As the base-case, we take a $1 \times 1$ "matrix" $(1)$. The eigenvalue of this matrix is unity. Next, we assume that the claim holds for $m \times m$ matrices with rows of all zeros except for a single one. Suppose we have a $(m+1) \times (m+1)$ matrix $T'$ of the same structure. If $T'$ is a permutation matrix, then we know its eigenvalues are roots of unity (Artin, 2011), so suppose that $T'$ is not a permutation matrix. Then $T'$ must have at least one columns that is all zeros. We then examine the characteristic polynomial $\phi(T')$. Without loss of generality, assume that column is the first column of the matrix. Then, we can write out the expression for the characteristic polynomial $\phi(T')$:

$$\phi(T') = \det(T' - \lambda I) = \det\left(\begin{array}{c|c} -\lambda & \cdots \\ \hline 0 & T'_{2:,2:} - \lambda I \end{array}\right) = -\lambda \cdot \det(T'_{2:,2:} - \lambda I) = -\lambda \cdot \phi(T'_{2:,2:})$$

where $T'_{2:,2:}$ is the $m \times m$ submatrix of $T'$ that omits the first row and column of $T'$. This submatrix is also a matrix with all zeros for every row except for a single one, since we eliminated a column of only zeros from $T'$. Thus, we know that the eigenvalues of $T'$ are 0 and the eigenvalues of $T'_{2:,2:}$, which, by the induction hypothesis, are also zero and roots of unity. $\qquad\square$

## B ADDITIONAL ALGORITHMIC DETAILS

### B.1 PARAMETERS INTRODUCED FOR FINITE DATA

Our derivations so far have assumed to be in the asymptotic regime where we have made perfect estimates of the Hankel matrix. In practice, with finite data, we only have the empirical Hankel matrix, $\hat{\mathcal{H}}$, which is subject to random perturbations. Naturally, some adjustments to the calculations

expressed in the previous section must be made to account for estimation error. There are three operations where estimation error influences the learning procedure: rank estimation via truncated SVD, determining full rank transition matrices $M^a$, and obtaining the observability partition to compute the random matrix $R$ in Sec. 4.3. For the Hankel matrix rank, we find that introducing a threshold on the low-rank approximation's reciprocal condition number to be sufficient. To test for transition full-rankness, we found that a minimum singular value $\sigma_{\min}$ threshold was acceptable. To find the full-rank observability partition, we consider two observation distributions to be equivalent if their $L^1$ norm falls below a threshold $\tau_{obs}$.

While the transition and observation likelihoods computed from the data will converge to the values true values asymptotically, approximation error prevents us from directly reading the parameter estimates as probabilities (Guo et al., 2016; Azizzadenesheli et al., 2016). Before using the learned model we project all parameters back to probability distributions via quadratic programming by minimizing the $L_2$ norm.

Algorithm pseudocode can be found in Alg. 1. We note that for all experiments, while it is theoretically correct to construct a block-diagonal rotation matrix $R$ by the full-rank observability partition as stated above, we find in practice it is sufficient to multiply by fully-dense random rotation matrix $R'$ after computing the initial SVD (line 4). This modification uses $AR'$ and $R'^T V$ to compute the linear PSR and takes $R = I$ instead at line 19. We still require a $\tau_{obs}$ parameter to compute the partition-level transition errors in Figures 3 and 4.

---

**Algorithm 1** Learn-POMDP

---

**Require:** Dataset $\mathcal{D} = (a_1 o_1, a_2 o_2 \dots)$, reciprocal cond. number $1/\kappa$, substring length $L$, minimum trans. mat singular value $\sigma_{min}$, observation sim. threshold $\tau_{obs}$:

1: $substrings \leftarrow \{(a_i o_i a_{i+1} o_{i+1} \dots a_{i+k} o_{i+k})\}_{i=1}^{|\mathcal{D}|/2-L}$
2: $\mathcal{H} \leftarrow \text{EstimateHankel}(substrings)$           $\triangleright$ Entries estimated via Eq. 6.
3: $U, \Sigma, V^T \leftarrow \text{TruncatedSVD}(\mathcal{H}, r, 1/\kappa)$
4: $A \leftarrow U\Sigma$
5: $m_0, \{M^{ao} : a \in \mathcal{A}, o \in \mathcal{O}\}, m_\infty \leftarrow \text{ComputePSR}(A, V^T, \mathcal{H})$     $\triangleright$ via Eqs. 7-9.
6: $M_{obs} = []$
7: **for** $a \in \mathcal{A}$ **do**
8:      $M^a \leftarrow \sum_{o \in \mathcal{O}} M^{ao}$
9:      **if** $\text{MinSingularValue}(M^a) > \sigma_{min}$ **then**        $\triangleright$ Detect full-rank actions.
10:         **for** $o \in \mathcal{O}$ **do**
11:            $(M_{obs}).\text{append}(M^{ao}(M^a)^{-1})$
12: $w_1, \dots w_{|M_{obs}|} \sim \text{Unif}(\mathbb{S}^{|M_{obs}|-1})$
13: $P' \leftarrow \text{Eigenvectors}(\sum_{i=1}^{|M_{obs}|} w_i (M_{obs})_i)$    $\triangleright$ via Eq. 18. Eigenvectors form columns of $P'$.
14: **for** $M^{ao}(M^a)^{-1} \in M_{obs}$ **do**
15:      $O^{ao} \leftarrow P'^{-1} M^{ao}(M^a)^{-1} P'$
16: $[S_1, \dots, S_k] \leftarrow \text{DetectPartitions}(\{O^{ao}\}, \tau_{obs})$        $\triangleright$ via procedure in Sec. B.1.
17: **for** $S_i \in [S_1, \dots, S_k]$ **do**
18:      $R_i \sim \text{Unif}(SO(|S_i|))$
19: $R \leftarrow \text{BlockDiag}([R_1, \dots, R_k], [S_1, \dots, S_k])$
20: $\tilde{P} \leftarrow P' R \, \text{diag}(R^T P'^{-1} m_\infty)$      $\triangleright$ Blocks are specified by indices in partitions $S_1, \dots, S_k$.
21: $\tilde{b} \leftarrow m_0 \tilde{P}$
22: **for** $(a, o) \in \mathcal{A} \times \mathcal{O}$ **do**
23:      $\tilde{O}^{ao} \tilde{T}^a \leftarrow \tilde{P} M^{ao} \tilde{P}^{-1}$
**Ensure:** $\tilde{b}, \{\tilde{O}^{ao} \tilde{T}^a : a \in \mathcal{A}, o \in \mathcal{O}\}$

---

### B.2 Runtime Complexity in Floating-Point Operations

The runtime of our approach, which we measure in floating-point operations, is dominated by the rank factorization of the Hankel matrix and computation of the PSR update matrices. We define the *full-observability length* of a POMDP (and notate as $n^{obs}$) to be the smallest length of histories and tests so that the Hankel matrix, whose rows and columns are indexed by action-

observation sequences enumerated up to this length, is full-rank. Suppose we are given a Hankel matrix that enumerates histories up to $n^{obs} + 1$ in the rows (so that $A_{hists^{-ao},:}$ in Eq. 7 is full-rank) and tests up to length $n^{obs}$ in the columns. The size of the Hankel matrix, then, must be $O((|\mathcal{A}||\mathcal{O}|)^{n^{obs}+1}) \times O((|\mathcal{A}||\mathcal{O}|)^{n^{obs}+1})$. Computing the truncated SVD with an appropriately set singular value, threshold, then, has runtime $O(|\mathcal{S}| \cdot (|\mathcal{A}||\mathcal{O}|)^{2(n^{obs}+1)})$. To compute the PSR update matrices, we pseudoinvert the right rank factor once, which also has complexity $O(|S| \cdot (|\mathcal{A}||\mathcal{O}|)^{2(n^{obs}+1)})$. Then, to compute each $M^{ao}$, we must pseudoinvert the right factor $A_{hists^{-ao},:}$, which has runtime $O(|\mathcal{S}| \cdot (|\mathcal{A}||\mathcal{O}|)^{2n^{obs}})$, and then compute the product $A_{hists^{-ao},:}^{\dagger} \mathcal{H}_{hists^{ao},:}(V^T)^{\dagger}$, which has runtime

$$O\left(|\mathcal{S}| \cdot (|\mathcal{A}||\mathcal{O}|)^{2n^{obs}+1}\right) + O\left(|\mathcal{S}|^2(|\mathcal{A}||\mathcal{O}|)^{n^{obs}+1}\right).$$

Putting everything together, we have a full runtime of:

$$O\left(|\mathcal{S}|(|\mathcal{A}||\mathcal{O}|)^{2(n^{obs}+1)} + |\mathcal{S}|^2(|\mathcal{A}||\mathcal{O}|)^{n^{obs}+2}\right) \tag{26}$$

Interestingly, our calculation suggests a runtime that is polynomial when the observability length $n^{obs}$ is known to be bounded by a constant. Further investigation to prove fixed-parameter tractability under a computational learning framework (e.g. PAC-learning) would be an interesting direction of future work.

## C  ADDITIONAL EXPERIMENTAL DETAILS

### C.1  ALGORITHM PARAMETER SELECTION

As discussed in Appendix B, the behavior performance of our learning algorithm depends on a few manually-specified parameters. This section reviews all the parameters that must be specified to run our approach, and the parameters values selected for our experiments (for a summery, see Table 1).

For Hankel estimation, of practical concern is the selection of the size Hankel matrix to estimate, or the sequences to include as row and column indices. Like many other approaches (Hsu et al., 2012; Balle et al., 2014), we that we use every possible action-observation sequence up to a certain length. We expose this length as an algorithm parameter. While smaller lengths will result in faster convergence of matrix entry estimates, selecting a length that is too short may result in a Hankel matrix whose approximate rank is strictly less than the number of states of the system. Our chosen lengths are included in Table 1. Automatically determining the proper Hankel size is an open question since the development of spectral approaches for PSRs (Wolfe et al., 2005; Boots et al., 2011; Balle et al., 2014), and remains an interesting question for future work.

The main parameter associated with learning PSRs is centered around the number of singular components to be used when computing the rank factorization of the Hankel matrix (Section 3.2). As mentioned in Section B.1, the main way to do this is by specifying a lower threshold on the empirical Hankel matrix lower rank approximation's reciprocal condition number. Empirically, we observe that empirical Hankel matrices tend to become more singular as the amount of data used to estimate them increases (Fig. 5). While any sufficiently small positive threshold may work with large amounts of data, in practice, larger thresholds will more quickly identify the number of states, at risk of omitting states. Of practical concern is the maximum number of singular values to compute to avoid computing an SVD of the *entire* Hankel matrix. The specified values for our experiments are shown in Table 1 under $1/\kappa$ and 'No. SVD,' respectively.

When recovering the observation distributions and partition-level transitions, we must specify thresholds to determine transition matrix full-rankness and a threshold that determines when observation distributions are similar (Section B.1). Inverting a near-singular matrix is highly undesirable when computing the matrices to joint-diagonalize in Eq. 17, so we specify conservatively high threshold on the smallest singular value on the smallest singular value $\sigma_{min}$ of the transition matrix. As discussed in Appendix B.1, in practice, a threshold $\tau_{obs}$ is not required to compute a random block-diagonal rotation matrix $R$. However, $\tau_{obs}$ is still required to compute partition-level transition likelihoods errors reported in Figures 3 and 4. We set a conservativitely high threshold $\tau_{obs}$ to merge observation distributions aggressively when plotting those figures.

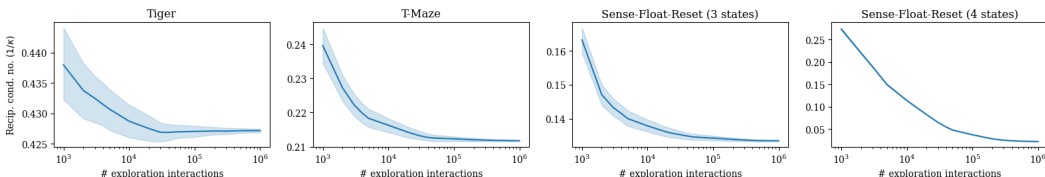

Figure 5: The ratio of the $r$th condition number over the largest condition number of Hankel matrices as the amount of observed data increases, where $r = |S|$ is the number of states of the POMDP. Each plot is averaged over 100 runs. The matrices become *more* singular as the amount of data increases. The sizes of the Hankel matrix correspond with Table 1.

Table 1: Chosen parameters for each domain in planning experiments described in Section 5, Fig. 3. Up and down arrows indicate a upper or lower threshold, respectively. The tuple reported for maximum indexing sequence lengths is ordered: (rows, columns).

|  | $1/\kappa \downarrow$ | $\sigma_{min} \uparrow$ | $\mathcal{H}$ max. seq. len. $\uparrow$ | No. SVD $\uparrow$ | $\tau_{obs} \uparrow$ |
|---|---|---|---|---|---|
| Tiger | 0.34 | 0.1 | (2, 1) | 20 | 0.1 |
| T-Maze | 0.1 | 0.01 | (2, 1) | 20 | 0.1 |
| Sense-Float-Reset (3 states) | 0.1 | 0.1 | (3, 2) | 20 | 0.1 |
| Sense-Float-Reset (4 states) | 0.015 | 0.1 | (4, 3) | 20 | 0.5 |

All PO-UCT planners require specification of a upper-confidence bound (UCB) constant to balance exploiting current estimated action value and exploring new actions. For all experiments, we use a UCB constant of $c = 2$. Our planners also limits all searches to a depth of three, and performs a fixed 1000 simulations per planning step.

## C.2    SENSITIVITY ANALYSIS AND WALL-CLOCK RUNTIME ESTIMATES

Additionally, we have included an analysis on the sensitivity of the truncated SVD step to both Hankel size and rank tolerance $1/\kappa$ (see Appendix C.1 for a description on parameters). As experiments results in Fig. 3 suggest, convergence of the number of states is a key step to the convergence of the overall algorithm.

Table 2 reports the number of estimated states against a variety of Hankel sizes and rank tolerances (and the rank of the Hankel matrix in limit of infinite data). Estimates of observation and transition likelihood error can be found in Tables 3 and 4 respectively. Runtimes can be found in Table 5. Maximum Hankel size was determined to be the largest to allow for a RAM allocation of under 64Gb and reasonable runtime of eight cores allocated on an Intel Xeon Gold 6140 CPU on a shared cluster. We observe that while more aggressive rank thresholds may arrive at the correct estimate with less data, they may also lead to an underestimation of the number of states. Lower thresholds will more likely underestimate the number of states, and require more data before the correct estimate is reached. Furthermore, larger Hankel sizes are required to estimate POMDPs with larger states, which tend to have longer observability lengths (Appendix B.2). For example, a Hankel size that enumerates histories of length four and tests of length three cannot fully capture a 14-state T-Maze. As Hankel size increases, so do the acceptable thresholds to estimate the number of states. Transition and observation likelihood errors, however, appear to be less influenced by Hankel size. These results suggest that the largest possible Hankel accomodated by running time and memory should be used for ease of selection of the remaining algorithm parameters.

We have also taken runtime estimates of the runtime of each component of the learning algorithm and PO-UCT search for the results reported in Figure 3, which we have included in Table 6. The algorithms have been implemented as unoptimized Python code running on two cores allocated from an Intel Xeon Gold 6140 CPU and 4Gb RAM on a shared cluster. We observe the most expensive part of the algorithm is the estimation of the Hankel matrix, because its length and width scales exponentially as we extend the maximum length of enumerated histories and tests (Appendix B.2). The remaining learning components of the algorithm can be highly vectorized, and when learning POMDPs of with small numbers of states (fewer than four states), their runtimes are relatively fast.

Table 2: Sensitivity analysis on the **estimated rank** of the Hankel matrix based chosen Hankel sizes and rank tolerances estimated from $10^7$ interactions in T-Maze environments with varying numbers of states. Hankel size is represented in the maximum lengths of action-observation sequences used to index the row and columns of the Hankel matrix, respectively. Rank tolerance is specified as $1/\kappa$, as discussed in Appendix C.1. Results are reported up to two significant figures, with trailing zeros truncated for space. Hankel rank for the corresponding Hankel size in the limit of infinite data is included in the GT column. Results reported are mean and standard deviation of the number of estimated states, aggregated over 20 seeds. A value of 'nan' is reported when no full-rank actions were found.

| $n$ states | $1/\kappa$ $\mathcal{H}$ seq. len. | GT | 1e-01 | 1e-02 | 1e-03 | 1e-04 | 1e-05 | 1e-06 |
|---|---|---|---|---|---|---|---|---|
| 4 | (2, 1) | 4 | $4 \pm 0$ | $4 \pm 0$ | $6 \pm .74$ | $13 \pm .73$ | $14 \pm .46$ | $14 \pm 0$ |
| | (3, 2) | 4 | $4 \pm 0$ | $4 \pm 0$ | $20 \pm 0$ | $20 \pm 0$ | $20 \pm 0$ | $20 \pm 0$ |
| | (4, 3) | 4 | $4 \pm 0$ | $4 \pm 0$ | $20 \pm 0$ | $20 \pm 0$ | $20 \pm 0$ | $20 \pm 0$ |
| 6 | (2, 1) | 5 | $4 \pm 0$ | $5 \pm 0$ | $7.9 \pm .94$ | $16 \pm .73$ | $18 \pm .4$ | $18 \pm 0$ |
| | (3, 2) | 6 | $5 \pm 0$ | $6 \pm 0$ | $20 \pm 0$ | $20 \pm 0$ | $20 \pm 0$ | $20 \pm 0$ |
| | (4, 3) | 6 | $6 \pm 0$ | $6 \pm 0$ | $20 \pm 0$ | $20 \pm 0$ | $20 \pm 0$ | $20 \pm 0$ |
| 8 | (2, 1) | 6 | $5 \pm 0$ | $6 \pm 0$ | $10 \pm 1.1$ | $20 \pm .3$ | $20 \pm 0$ | $20 \pm 0$ |
| | (3, 2) | 8 | $6 \pm 0$ | $8 \pm 0$ | $20 \pm 0$ | $20 \pm 0$ | $20 \pm 0$ | $20 \pm 0$ |
| | (4, 3) | 8 | $6 \pm 0$ | $8 \pm 0$ | $20 \pm 0$ | $20 \pm 0$ | $20 \pm 0$ | $20 \pm 0$ |
| 10 | (2, 1) | 7 | $6 \pm 0$ | $7 \pm 0$ | $13 \pm 1.4$ | $20 \pm 0$ | $20 \pm 0$ | $20 \pm 0$ |
| | (3, 2) | 9 | $6 \pm 0$ | $8 \pm 0$ | $20 \pm 0$ | $20 \pm 0$ | $20 \pm 0$ | $20 \pm 0$ |
| | (4, 3) | 10 | $7 \pm 0$ | $10 \pm .3$ | $20 \pm 0$ | $20 \pm 0$ | $20 \pm 0$ | $20 \pm 0$ |
| 12 | (2, 1) | 8 | $7 \pm 0$ | $8 \pm 0$ | $17 \pm .93$ | $20 \pm 0$ | $20 \pm 0$ | $20 \pm 0$ |
| | (3, 2) | 10 | $7 \pm 0$ | $9 \pm 0$ | $20 \pm 0$ | $20 \pm 0$ | $20 \pm 0$ | $20 \pm 0$ |
| | (4, 3) | 12 | $8 \pm 0$ | $14 \pm 1.2$ | $20 \pm 0$ | $20 \pm 0$ | $20 \pm 0$ | $20 \pm 0$ |
| 14 | (2, 1) | 9 | $8 \pm 0$ | $9 \pm 0$ | $20 \pm .57$ | $20 \pm 0$ | $20 \pm 0$ | $20 \pm 0$ |
| | (3, 2) | 11 | $8 \pm 0$ | $10 \pm 0$ | $20 \pm 0$ | $20 \pm 0$ | $20 \pm 0$ | $20 \pm 0$ |
| | (4, 3) | 13 | $9 \pm 0$ | $20 \pm 0$ | $20 \pm 0$ | $20 \pm 0$ | $20 \pm 0$ | $20 \pm 0$ |

Table 3: Sensitivity analysis on the **observation error** (in $L_1$ norm) associated with Table 2. Estimates are only taken when the number of estimated states is equivalent to the ground truth POMDP. For all experiments the full-rank transition threshold $\sigma_{min}$ is set to 0.01. A value of 'nan' is reported when no full-rank actions were found. If no standard deviation is included, only one seed of twenty succeeded in finding a full-rank action.

| $n$ states | $1/\kappa$ $\mathcal{H}$ seq. len. | 1e-01 | 1e-02 | 1e-03 | 1e-04 | 1e-05 | 1e-06 |
|---|---|---|---|---|---|---|---|
| 4 | (2, 1) | $0.09 \pm 0.2$ | $0.11 \pm 0.28$ | nan | nan | nan | nan |
| | (3, 2) | $0.031 \pm 0.012$ | $0.033 \pm 0.015$ | nan | nan | nan | nan |
| | (4, 3) | $0.026 \pm 0.011$ | $0.025 \pm 0.012$ | nan | nan | nan | nan |
| 6 | (2, 1) | nan | nan | 0.47 | nan | nan | nan |
| | (3, 2) | nan | $0.2 \pm 0.45$ | nan | nan | nan | nan |
| | (4, 3) | $0.063 \pm 0.025$ | $0.077 \pm 0.054$ | nan | nan | nan | nan |
| 8 | (2, 1) | nan | nan | nan | nan | nan | nan |
| | (3, 2) | nan | $0.38 \pm 0.24$ | nan | nan | nan | nan |
| | (4, 3) | nan | $0.59 \pm 1.5$ | nan | nan | nan | nan |
| 10 | (2, 1) | nan | nan | nan | nan | nan | nan |
| | (3, 2) | nan | nan | nan | nan | nan | nan |
| | (4, 3) | nan | $0.29 \pm 0.13$ | nan | nan | nan | nan |
| 12 | (2, 1) | nan | nan | nan | nan | nan | nan |
| | (3, 2) | nan | nan | nan | nan | nan | nan |
| | (4, 3) | nan | $1.7 \pm 0.017$ | nan | nan | nan | nan |
| 14 | (2, 1) | nan | nan | nan | nan | nan | nan |
| | (3, 2) | nan | nan | nan | nan | nan | nan |
| | (4, 3) | nan | nan | nan | nan | nan | nan |

Table 4: Sensitivity analysis on the **transition error** (in $L_1$ norm) associated with Table 2. Values are reported using the same estimation protocol as Table 3, except transition likelihoods were measured.

| $n$ states | $1/\kappa$ $\mathcal{H}$ seq. len. | 1e-01 | 1e-02 | 1e-03 | 1e-04 | 1e-05 | 1e-06 |
|---|---|---|---|---|---|---|---|
| 4 | (2, 1) | $0.095 \pm 0.32$ | $0.067 \pm 0.21$ | nan | nan | nan | nan |
|   | (3, 2) | $0.018 \pm 0.015$ | $0.028 \pm 0.045$ | nan | nan | nan | nan |
|   | (4, 3) | $0.017 \pm 0.022$ | $0.011 \pm 0.0077$ | nan | nan | nan | nan |
| 6 | (2, 1) | nan | nan | 0.46 | nan | nan | nan |
|   | (3, 2) | nan | $0.073 \pm 0.13$ | nan | nan | nan | nan |
|   | (4, 3) | $0.021 \pm 0.014$ | $0.06 \pm 0.091$ | nan | nan | nan | nan |
| 8 | (2, 1) | nan | nan | nan | nan | nan | nan |
|   | (3, 2) | nan | $0.033 \pm 0.024$ | nan | nan | nan | nan |
|   | (4, 3) | nan | $0.13 \pm 0.36$ | nan | nan | nan | nan |
| 10 | (2, 1) | nan | nan | nan | nan | nan | nan |
|   | (3, 2) | nan | nan | nan | nan | nan | nan |
|   | (4, 3) | nan | $0.047 \pm 0.038$ | nan | nan | nan | nan |
| 12 | (2, 1) | nan | nan | nan | nan | nan | nan |
|   | (3, 2) | nan | nan | nan | nan | nan | nan |
|   | (4, 3) | nan | $0.23 \pm 0.0087$ | nan | nan | nan | nan |
| 14 | (2, 1) | nan | nan | nan | nan | nan | nan |
|   | (3, 2) | nan | nan | nan | nan | nan | nan |
|   | (4, 3) | nan | nan | nan | nan | nan | nan |

Table 5: **Runtime estimates** for sensitivity analysis on the T-Maze environments shown in Table 2. All Hankel matrices were estimated at maximum size using sparse representations and then indexed to form smaller Hankel matrices, which is why there is little variation in runtime across T-Maze instances with varying numbers of states. All estimates are reported as mean and standard deviation over 20 seeds.

| $\mathcal{H}$ seq. len. | $n$ states | PSR (s) | POMDP (s) |
|---|---|---|---|
| (2, 1) | 4 | $0.19 \pm 0.098$ | $0.2 \pm 0.099$ |
|   | 6 | $0.24 \pm 0.12$ | $0.25 \pm 0.12$ |
|   | 8 | $0.25 \pm 0.11$ | $0.25 \pm 0.11$ |
|   | 10 | $0.29 \pm 0.12$ | $0.29 \pm 0.12$ |
|   | 12 | $0.33 \pm 0.15$ | $0.34 \pm 0.16$ |
|   | 14 | $0.35 \pm 0.13$ | $0.36 \pm 0.13$ |
| (3, 2) | 4 | $1.1 \pm 0.41$ | $1.1 \pm 0.41$ |
|   | 6 | $1.6 \pm 0.54$ | $1.6 \pm 0.54$ |
|   | 8 | $2.3 \pm 0.77$ | $2.3 \pm 0.77$ |
|   | 10 | $3.1 \pm 0.93$ | $3.1 \pm 0.93$ |
|   | 12 | $3.9 \pm 1.3$ | $3.9 \pm 1.3$ |
|   | 14 | $6.1 \pm 13$ | $6.1 \pm 13$ |
| (4, 3) | 4 | $77 \pm 32$ | $77 \pm 32$ |
|   | 6 | $140 \pm 33$ | $140 \pm 33$ |
|   | 8 | $250 \pm 75$ | $250 \pm 75$ |
|   | 10 | $450 \pm 130$ | $450 \pm 130$ |
|   | 12 | $740 \pm 220$ | $740 \pm 220$ |
|   | 14 | $1.0e+3 \pm 260$ | $1.0e+3 \pm 260$ |

| $n$ states | $\mathcal{H}$ estim. time (s) |
|---|---|
| 4 | $910 \pm 210$ |
| 6 | $930 \pm 190$ |
| 8 | $960 \pm 220$ |
| 10 | $980 \pm 220$ |
| 12 | $970 \pm 220$ |
| 14 | $920 \pm 200$ |

Table 6: Runtime estimates from Fig. 3. PSR and POMDP columns are the total estimated times for learning each model, respectively. The last column describes the average planning time per planning step. The entries report mean and standard deviations, in seconds, up to two significant figures. Hankel estimates are the total amount of time to evaluate Eq. 6 on $10^6$ interactions.

|                | $\mathcal{H}$ estim. (s) | PSR (s)          | POMDP (s)         | EM (s)      | Plan. (s/step) |
| -------------- | ------------------------ | ---------------- | ----------------- | ----------- | -------------- |
| Tiger          | $16 \pm 8.6$             | $0.013 \pm .0061$ | $0.015 \pm 0.0068$ | $3.7 \pm 2.4$ | $3.4 \pm 1.2$  |
| T-Maze         | $18 \pm 12$              | $0.022 \pm 0.041$ | $0.024 \pm 0.015$ | $5.7 \pm 4.2$ | $4.1 \pm 2$    |
| SFR (3 states) | $27 \pm 16$              | $0.017 \pm 0.011$ | $0.019 \pm 0.011$ | $15 \pm 10$ | $3.5 \pm 1.4$  |
| SFR (4 states) | $83 \pm 28$              | $0.29 \pm 0.14$  | $0.29 \pm 0.14$   | $350 \pm 250$ | $4.7 \pm 1.6$  |

### C.3 PLANNING PERFORMANCE OF DIFFERENT SAMPLING AND DISTRIBUTION ROUNDING STRATEGIES

There are many ways to sample action-observation trajectories when deriving a UCT-based search algorithm for planning on POMDPs. As discussed by Silver & Veness (2010), there are largely two approaches, which differ in how the latent state treated as the search propogates down a branch of the search tree.

1. Upon choosing an action, propogate the full belief state using a process update (e.g. multiplying by $\tilde{T}^a$ of Theorem 1). Compute the mixture observation distribution weighted by that belief state, from which we can sample the emitted observation.

2. Upon choosing an action, sample a *single* latent state (or observability partition, in the context of Theorem 1). Look up the observation distribution associated with the action and sampled state, and then sample the observation.

Silver & Veness (2010, Lemma 2) prove that the observation distribution under these two sampling strategies are equivalent, so a UCT-based search will perform the same using either approach. They also argue the latter is more computationally efficient for systems with a large number of states.

UCT-based search algorithm may only use one or some variant of both approaches when planning with the models learned by the algorithms discussed in this paper. Because PSRs do not yield explicit transition likelihood estimates, we do not have the state distributions used to sample individual states for the second approach. The first approach, however, can still be applied. Given a PSR sufficient statistic $m$, we compute products $m \cdot M^{ao} \cdot m_\infty$ for the chosen action and all possible observations, yielding the observation likelihoods. The learned partition-level POMDPs may apply the first approach in the same way. Furthermore, the second approach may be applied to the learned partition-level POMDPs by sampling the next *observability partition*, rather than state. Given a partition-level belief $\tilde{b}$, we first compute the partition-level belief distribution by summing across across appropriate indices, and then sampling the current partition $S$. We can then compute the *conditional* observation distribution by first computing the *conditional* partition-level belief vector

$$\tilde{b}_S = \frac{\tilde{b} \otimes \mathbb{I}_S}{(\tilde{b} \otimes \mathbb{I}_S)^T \cdot \mathbf{1}}$$

where $\mathbb{I}_S$ is a vector with entries of value one for indices in partition $S$ and zero otherwise, $a$ is the selected action by the search algorithm, and $\otimes$ the element-wise product. The conditional observation distribution is then found by computing products $\tilde{b}_S \tilde{T}^a O^{ao}$ for all observations $o \in \mathcal{O}$, and projecting the distribution to deal with approximation error as handlied in the first appraoch. Verifying the correctness of this calculation is a straightforward extension of the proof of Theorem 1.

Our approach to handling rounding estimated likelihoods to proper probability distribution parameters is different across the two sampling strategies. When planning using the first sampling approach, we first compute the estimated observation distribution, project the distribution, and then sample. When planning with the second, we compute the estimated partition-level likelihoods, project the distribution, sample a *partition*, and then sample the appropriate observation. In practice, we do not observe an empirical difference between these sampling and rounding approaches for planning with rewards learned as observations.

### C.4 SLOWER CONVERGENCE OF TRANSITION LIKELIHOODS

A common approach to convergence analysis of tensor decomposition methods would first argue the convergences of the SVD of our Hankel matrix and *then* argue convergence of the eigendecompositions of the matrix discussed in 18 (Moitra, 2018). PSRs only depend on the convergence of the SVD, while our learned POMDPs depends on the convergence of both the SVD and eigendecomposition. In our reward-specification experiments (Fig. 4), accurate transitions are required to correctly assign reward to the desired goal state. The slow convergence of performance of the planner in the directional hallway environment suggests that more data is required to obtain accurate likelihood estimates of transition and observation matrices.

### C.5 EXPERIMENTAL DOMAINS

Here, we document any environment we have modified, or any novel environments we introduced in this work. We have used the original Tiger domain as described by Kaelbling et al. (1998), which we omit from our discussion below.

#### C.5.1 SENSE-FLOAT-RESET

As discussed in Sec. 4, the transition dynamics and observation emissions of Sense-Float-Reset are the same as those of Float-Reset introduced by Littman & Sutton (2001), but augment the system with a passive sensing action.

**Transition dynamics**. In an $n$ state float-reset problem, the 'reset' state is typically denoted as $s^0$ and the remaining states $\{s^1, s^2, \ldots, s^{n-1}\}$. The `float` action allows the system to translate to adjacent integer states (or loop at the ends):

$$P(s_{t+1} = s^j | s_t = s^i, a_t = \texttt{float}) = \begin{cases} 0.5, & i = j = 0, (n-1) \text{ or } i = j \pm 1, \\ 0 & \text{otherwise} \end{cases}$$
$$\forall i, j \in \{0, \ldots, n-1\}.$$

The `reset` action deterministically sets the state to $s^0$, e.g.

$$P(s_{t+1} = s^0 | s_i, a_t = \texttt{reset}) = 1 \quad \forall i \in \{0, \ldots, n-1\}.$$

The `sense` action does not change the state, e.g.

$$P(s_{t+1} = s^i | s_t = s^i, a_t = \texttt{sense}).$$

**Observation emissions**. The `float` action only emits an observation of zero, e.g.

$$P(o_t = 0 | s_t = s^i, a_t = \texttt{float}) = 1 \quad \forall i \in \{0, \ldots, n-1\}.$$

The `reset` and `sense` actions emit a `1` when $s_t$ is in $s^0$ (the 'reset state'), and `0` otherwise:

$$P(o_t = 1 | s_t = s^i, a_t = \texttt{reset}) = P(o_t = 1 | s_t = s^i, a_t = \texttt{sense})$$
$$= \begin{cases} 1, & i = 0, \\ 0, & \text{otherwise,} \end{cases}$$
$$P(o_t = 0 | s_t = s^i, a_t = \texttt{reset}) = P(o_t = 0 | s_t = s^i, a_t = \texttt{sense})$$
$$= \begin{cases} 1, & i \in \{1, \ldots, n-1\}, \\ 0, & \text{otherwise.} \end{cases}$$

**Reward function**. In all of our experiments, we specify a deterministic tabular reward of $+1$ when the system exists $s^1$, and emit a reward of $0$ otherwise,

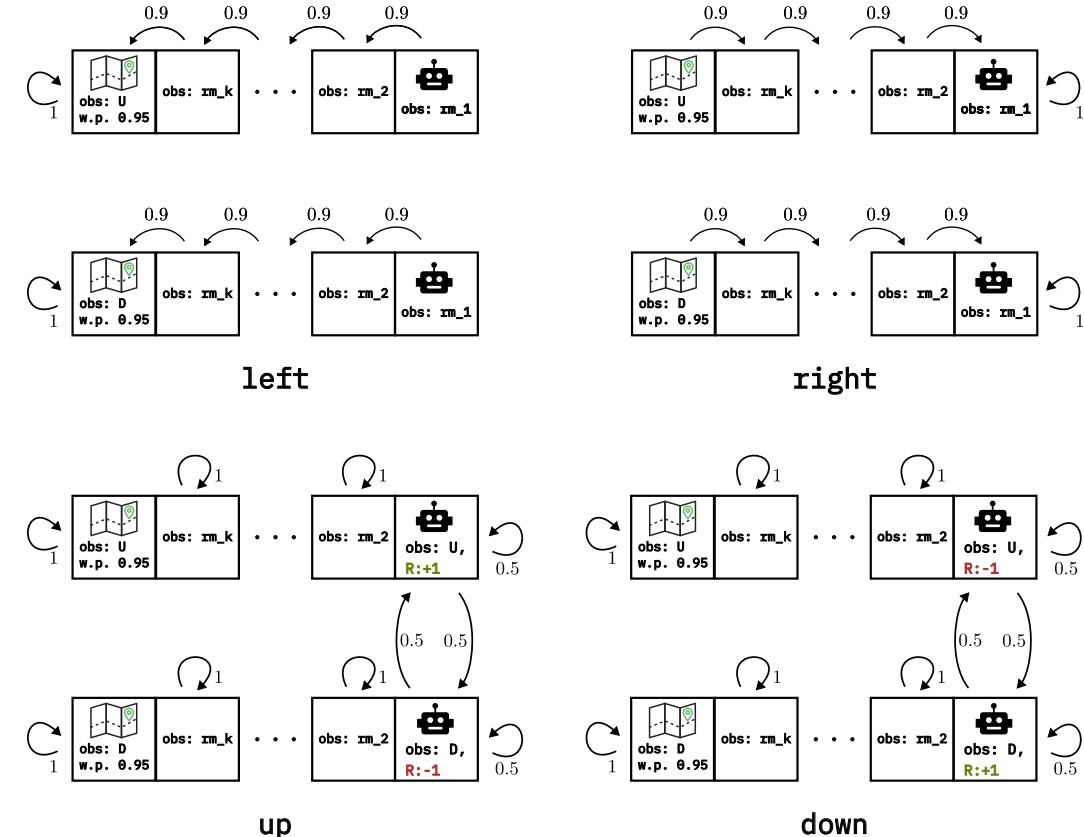

Figure 6: T-Maze dynamics and observation distributions. Edges are labeled with transition probabilities, and nonzero rewards are emitted deterministically from annotated states (and rewards of zero from non-annotated states). Self-loop edges with probability less than 1 are omitted. Leftmost 'map' states in the top hallway emit a U with probability with probability 0.95 and D with probability 0.05 (and vice-versa for the bottom hallway). All other observations are deterministic.

$$
\mathrm{P}(r_t = r | s_t = s^i, a_t = a) = \begin{cases} 1, & r = +1, i = 1 \text{ or } r = 0, i \in \{0, 2, \ldots n-1\} \\ 0, & \text{otherwise.} \end{cases}
$$

$$
\forall a \in \{\texttt{float}, \texttt{reset}, \texttt{sense}\}.
$$

### C.5.2 T-MAZE

We present a version of T-Maze similar to the one described by Allen et al. (2024). Since we allow actions to determine observation emissions (Allen et al. determine observations by states), our T-Maze POMDP has fewer states than their version. This environment is more easily explained pictorally than explicit probability expressions. See Fig. 6 for a depiction of transition dynamics and observation emissions. For the truncated T-Maze used for experiments in Section 5 Figure 3, the number of room states was set to $k = 1$.

### C.5.3 NOISY HALLWAYS

The transition dynamics common to *directional hallway* and *noisy hallway* can be found in Fig. 7. Across both domains, under the stay and reset actions, the environment will emit either end-left or end-right with probability 0.5. Furthermore, the left and the right states will emit end-left and end-right, respectively, with probability 0.8 under actions left, and will omit the incorrect observation (end-right from leftmost state, and vice-versa) with probability 0.2.

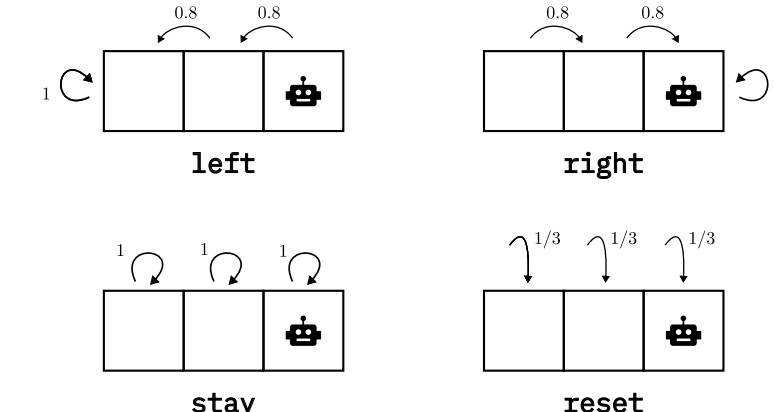

Figure 7: Noisy-hallway dynamics shared between *directional hallway* and *noisy hallway* domains. Edges are labeled with transition probabilities, and nonzero rewards are emitted deterministically from annotated states. Self-loop edges with probability less than 1 are omitted. For the `reset` action, the agent samples the next state uniformly from the available states.

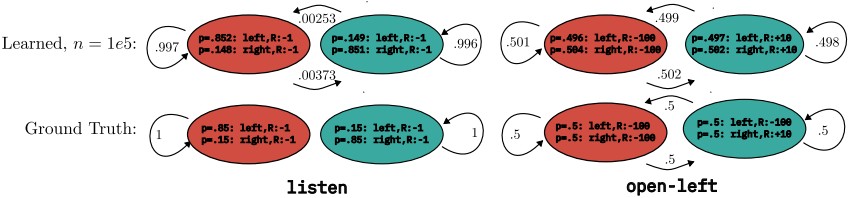

Figure 8: A comparison of a learned instance of Tiger after $10^5$ samples compared to the ground truth for the `listen` and `open-left` actions. For each action, nodes are annotated with their observation emission probabilities, and edges are annotated with their transition probabilities.

The two domains differ on the observation distribution of the middle state under the actions `left` and `right`. In the directional environment, under `left`, the observation `end-left` is emitted with probability 0.8, and the `end-right` emitted with probability 0.2. Similaritly, under the `right` action, the `end-right` observation is emitted with probability 0.8, and the `end-left` observation is emitted with probability 0.2. In the noisy environment, under both `left` and `right` actions, either `end-left` or `end-right` may be emitted with probability 0.5.

There is no reward function is given, since both of these experiments are used in the reward-specification experiments discussed in Sec. 5, Fig. 4. For the directional environment, the rewarded tuples are (`left`,`end-left`) and (`right`,`end-right`). For the noisy environment, the rewarded tuples are (`left`,`end-left`), (`right`,`end-right`), (`left`,`end-right`), and (`right`,`end-left`).

It is important to note that these domains are fully-recoverable by our algorithm, even though there are fewer observations than states. This is because all actions aside from `reset` are full-rank and that the observation *distributions* associated with these actions are distinct.

### C.6 EXAMPLE OUTPUT OF ALGORITHM

Here, we include an example of the learned model after estimates of transition and observation matrices have nearly converged. In Tiger, where each state has a unique observation distribution, the learned model, as illustrated in Fig. 8, shows close agreement between the learned and ground-truth transition and observation matrices. These results confirm that by estimating the similarity transform, we can recover the true observation and transition models.

# D USE OF LARGE LANGUAGE MODELS

Our use of large language models is solely limited to the completion of routine coding tasks in Python. These routine tasks consist of experiment launching, data retrieval infrastructure, and plotting code.

