# OpenReview forum: "Towards Learning POMDPs Without Full Observability"
_ICLR.cc/2026/Conference — Submitted to ICLR 2026_

### Official Review · Reviewer_iKCN · 2025-10-18

**Soundness:** 3
**Presentation:** 3
**Contribution:** 2
**Rating:** 4
**Confidence:** 2

**Summary:**

This work presents a method for learning the transition and observation functions of POMDPs from action-observation sequences obtained through random exploration. The method works by estimating a similarity transform $P$, converting a standard learned PSR into an equivalent basis where transition and observation probabilities can be directly recovered, enabling downstream inference and planning.

**Strengths:**

**Originality of the approach.** The paper presents a novel method for estimating the transition and observation functions of POMDPs.

**Presentation flow.** The paper is technically deep, and even quite dense at times. However, the ideas are presented in a way that benefits the reader by allowing them to build on each other.

**Theoretical Soundness.** The paper establishes the correctness of the approach under clear assumptions.

**Significance of the contribution** Though it may be somewhat limited to very small POMDPs and comes with significant scalability concerns, the ability of the method to recover the transition and observation likelihoods is a clear advantage over prior work.

**Weaknesses:**

**Scalability.** The method is constrained to very small-scale domains. The data and computation requirements are impractical for even modest state spaces and observation sequence lengths. The approach is only validated on tiny POMDPs with 3 or 4 states in the state space.

**Planning evaluation.** While the paper motivates its approach through planning, as stated above, the experimental domains are so small that it is unclear whether the claimed potential planning benefits would generalize to more realistic problems.

**Practical motivation.** The broader motivation and real-world relevance of learning full POMDP models through this method are questionable. Even if the method could be scaled considerably compared to how it works presently, I don't immediately see when such a computationally intensive approach would be preferable to modern function-approximation or model-free alternatives.

**Questions:**

* In the problem setting, the authors mention "For the purposes of evaluation, we may also require the agent to learn a tabular reward R function by including rewards as observations", but it's unclear whether this is done in the experiments section. Could the authors please clarify this point?
* Could the authors please elaborate on potential methods for scaling the approach?

Typos / etc.
* Line 245: "We call this alternate grouping is called a..."

---

> ### Author Response · Authors · 2025-12-03
> **Response to iKCN**
>
> Dear reviewer,
>
> Thank you for taking the time to provide valuable feedback. We address the points you have individually raised below:
>
> We address these points together:
> > The method is constrained to very small-scale domains. The data and computation requirements are impractical for even modest state spaces and observation sequence lengths. The approach is only validated on tiny POMDPs with 3 or 4 states in the state space.
>
> > Planning evaluation. While the paper motivates its approach through planning, as stated above, the experimental domains are so small that it is unclear whether the claimed potential planning benefits would generalize to more realistic problems.
>
> > Could the authors please elaborate on potential methods for scaling the approach?
>
>
>
> Please see our response to all reviews above in regard to environment scale. We stress that this is the first time we can provably recover certain toy POMDPs found in the literature.
>
> > Practical motivation. The broader motivation and real-world relevance of learning full POMDP models through this method are questionable. Even if the method could be scaled considerably compared to how it works presently, I don't immediately see when such a computationally intensive approach would be preferable to modern function-approximation or model-free alternatives.
>
> We believe the literature supports the value of model-based techniques in many (if not all) partially observable domains. For example, model-free RL methods require some form of memory to perform well in a given environment (or condition on the whole history of actions and observations, which cannot scale well) [1, 2].
> Transformers have been shown to be a poor choice to maintain some memory featurization because they are fixed computational circuits that do not contain any internal memory mechanism [3].
> The most performant methods include a latent state learned by an RNN as input to a policy trained by a model-free RL method.
> Successful applications of variants of this technique require access to the unobserved state at training time [2] or some score to measure the degree of non-Markov behavior of the underlying system [1].
>
> The RNN-augmented state used in RL-based methods works on a similar idea to PSRs, in which the recurrent signal behaves as the 'sufficient statistic' of the PSR. Thus, like PSRs, we cannot directly interpret explicit transition or observation likelihoods.
> In scenarios when we would like to learn the model *first* and then specify behavior later, RNN-augmented models may run into the reward-specification challenges as highlighted by our new reward-specification experiment in Section 5. Importantly, this challenge is especially true when the desired behavior is not known before training the model.
>
>
> > In the problem setting, the authors mention "For the purposes of evaluation, we may also require the agent to learn a tabular reward R function by including rewards as observations", but it's unclear whether this is done in the experiments section. Could the authors please clarify this point?
>
> Yes, we included tabular rewards for each environment. We have added a section in the appendix describing each domain and the reward functions used.
>
> **References**
>
> [1] C. Allen et al., “Mitigating Partial Observability in Sequential Decision Processes via the Lambda Discrepancy,” Advances in Neural Information Processing Systems, 2024.
>
> [2] A. Wang, A. C. Li, T. Q. Klassen, R. T. Icarte, and S. A. Mcilraith, “Learning Belief Representations for Partially Observable Deep RL,” in Proceedings of the 40th International Conference on Machine Learning, 2023.
>
> [3] C. Lu, R. Shi, Y. Liu, K. Hu, S. S. Du, and H. Xu, “Rethinking transformers in solving POMDPs,” in Proceedings of the 41st International Conference on Machine Learning, 2024.

---

### Official Review · Reviewer_MsTT · 2025-10-29

**Soundness:** 3
**Presentation:** 3
**Contribution:** 3
**Rating:** 6
**Confidence:** 3

**Summary:**

The paper studies learning discrete POMDP parameters from a single long action–observation sequence. Building on PSR/Hankel factorizations, the authors show how to compute a similarity transform that maps a linear PSR model to the original POMDP parameters (initial belief, observation, and transition matrices) under a set of assumptions. Because recovering each state separately is often infeasible, the authors prove recovery up to an observability partition, in which states that produce identical observation distributions across all full-rank actions are merged into the same group. Algorithmically, they (i) estimate a Hankel matrix from trajectories, (ii) obtain a low-rank factorization, (iii) recover a similarity transform via joint diagonalization over full-rank actions, and (iv) infer partition-level transitions and per-action observation matrices. Experiments on Tiger, T-Maze, and Sense-Float-Reset variants evaluate parameter recovery errors and planning returns with a PO-UCT planner run on the learned model.

**Strengths:**

- The paper introduces recovery up to a full-rank observability partition and a similarity-transform approach that reconstructs POMDP parameters via joint diagonalization over full-rank actions. This combination appears novel and relaxes identifiability assumptions of prior tensor-based POMDP methods
- The work moves PSR theory toward interpretable model-based planning, yielding explicit $b_\pi, O$, and $T$ parameters instead of opaque predictors. The partition perspective offers a practical middle ground between exact recovery and real-world observability, with potential impact on safe or explainable model-based RL.
- The empirical tests, though small, validate the feasibility of the end-to-end pipeline from estimation to planning.
- The Sense–Float–Reset example and the step-by-step reconstruction pipeline make the ideas concrete. Figures illustrating partitions and transitions are clear and help the reader follow the matrices involved.

**Weaknesses:**

- Several assumptions look strong for realistic POMDPs and probably require more discussion:
    1. the chain over $(s,a,o)$ under a uniform random policy is ergodic and mixes to a stationary distribution from which the Hankel is effectively estimated;
    2. existence and practical identification of “full-rank actions” for each state partition;
    3. independence properties of Back’s rows when restricted to histories; and
    4. starting the Hankel with the stationary distribution $b_{\pi}$ rather than an arbitrary $b_0$ (the paper later argues it emerges in the limit, but this places the burden on data).

    I would recommend a dedicated subsection up front explaining why such assumptions are acceptable, when these hold in common environments/benchmarks, and where they might fail.
- Only tiny POMDPs (3–6 states) are considered. Several plotted trends are not discussed in the text (e.g., when the proposed method underperforms baselines in planning return), and there is little analysis of why.
- A comparison to baselines sharing the same assumptions and goal (no access to the transition/observation dynamics+model learning) is missing (e.g., model-based RL).
- Although the appendix derives an exploration count (Figure 5), the text never reports total time (wall clock); the Hankel grows exponentially in the observability length, and PO-UCT is time-demanding, so practical limits matter.
- Planning is done with a shallow PO-UCT at the partition level (depth 3/1000 sim). It is unclear whether differences in returns stem from model quality or planner mismatch.
- Even if the actions are played uniformly at the model training time, there's no guarantee of _where_ the model is accurate after a finite number of steps; according to the underlying MDP dynamics, some parts of the MDP might be unexplored and so turn the model inaccurate in those regions, and there is no way to detect that phenomenon. As a result, the approach doesn't guarantee the usefulness of the learned model in a finite number of steps.

**Questions:**

- In Section 2, the observation model is written as $P(o_t = o \mid s_t = s, a_t = a)$  “on leaving the state,” This is non-standard against the usual $P(o_{t + 1} \mid s_{t + 1}, a_t)$ convention. How does this impact the belief computation? Are the two notions of observation functions strictly equivalent?
- In practice, you do not know which actions are full-rank. How do you detect $A_{\text{full}}$ from data robustly? What happens when no action is full-rank for some states?
- Could you expand on the practical plausibility of the Assumptions presented in Section 3.3? (cf. weaknesses)
- Does it make sense to start the computation from the stationary distribution $b_\pi$ (that you do not have access to since you can't observe the states)? Moreover, there is a stark difference between beliefs that you infer from sequences of actions and observations and the underlying occupancy measure of the (hidden) MDP. Could you elaborate on that?
- Is the theory expandable to non-uniform exploration strategies, e.g., with stochastic behavioral policies having full support (as standard in regularized MDPs)?

---

> ### Author Response · Authors · 2025-12-03
> **Response to MsTT (part 1/2)**
>
> Dear reviewer,
>
> Thank you for a careful evaluation of our paper!
>
> > Several assumptions look strong for realistic POMDPs and probably require more discussion:
>
> Please see the shared response to all reviewers for additional remarks on the restrictiveness of our assumptions, which we introduce in a new section (4.1.1).
>
> We address these points together:
> > existence and practical identification of “full-rank actions” for each state partition;
>
> > starting the Hankel with the stationary distribution rather than an arbitrary (the paper later argues it emerges in the limit, but this places the burden on data).
>
> > In practice, you do not know which actions are full-rank. How do you detect from data robustly? What happens when no action is full-rank for some states?
>
> Please see the shared response to all reviewers for an explanation of the practical existence of full-rank actions. As stated (and now emphasized) in the paper, we can check the existence of a full-rank action by computing the singular value decomposition of the matrix $M^a$ found in Equation 16.
>
> In our problem we can only take a *single* sample of $b_0$ at the beginning, so we cannot hope to recover $b_0$. However, there are alternate problem formulations that could allow for the possibility of learning $b_0$. One example is in an episodic context, where the environment is 'reset' after a certain number of actions (provided that the initial distribution supports all states in the system). Rather than estimating the Hankel from using a 'sliding window' along a single trajectory, frequency counts on episode trajectories could be done instead.
>
>
> > Only tiny POMDPs (3–6 states) are considered. Several plotted trends are not discussed in the text (e.g., when the proposed method underperforms baselines in planning return), and there is little analysis of why.
>
> We have included experimental results for runtimes and parameter choices in Appendix C.2. This was due to a bug in our PO-UCT planner, which has been fixed. We have updated Figure 3, which has a minimal performance gap between POMDPs and PSRs.
>
> > A comparison to baselines sharing the same assumptions and goal (no access to the transition/observation dynamics+model learning) is missing (e.g., model-based RL).
>
> Our paper is largely focused on the exploration and model-estimation part of the problem and the consequences for why having a model is useful. We have added baselines that many reviewers have raised that we feel are relevant in this problem setting. We are not innovating on learning a *policy*, which is why we adopt standard planning methods (PO-UCT planning). We agree that model-based RL can serve as an alternate way to validate the usability of the models, but we feel that the choice of what guides the behavior of the agent is arbitrary and orthogonal to the subject studied in this particular work.
>
> > Although the appendix derives an exploration count (Figure 5), the text never reports total time (wall clock); the Hankel grows exponentially in the observability length, and PO-UCT is time-demanding, so practical limits matter.
>
> We have included runtime estimates of planning and our model learning approach in Appendix C.2.
>
> > Planning is done with a shallow PO-UCT at the partition level (depth 3/1000 sim). It is unclear whether differences in returns stem from model quality or planner mismatch.
>
> Due to the size of POMDPs used for evaluation, the reward signals are not very sparse. In all problem instances, the agent can obtain some reward signal within 3 steps to guide planning. The quality of that reward signal will depend on the efficacy of the learned model.
>
>
> > Even if the actions are played uniformly at the model training time, there's no guarantee of where the model is accurate after a finite number of steps; according to the underlying MDP dynamics, some parts of the MDP might be unexplored and so turn the model inaccurate in those regions, and there is no way to detect that phenomenon. As a result, the approach doesn't guarantee the usefulness of the learned model in a finite number of steps.
>
> We agree that our theoretical claim is found in the limit of infinite data, after which the visitation distribution over states has converged to a stationary distribution. To prove a finite-sample guarantee, you correctly point out that the sample complexity of our approach will likely depend on the "mixing time" of the Markov chain $(s_t, a_t, o_t)$ to converge to a stationary distribution. As stated in Appendix A.5, this type of analysis is beyond the scope of the current paper, and we look forward to completing it in future work.

---

> ### Author Response · Authors · 2025-12-03
> **Response to MsTT (part 2/2)**
>
> > In Section 2, the observation model is written as
>  “on leaving the state,” This is non-standard against the usual
>  convention. How does this impact the belief computation? Are the two notions of observation functions strictly equivalent?
>
> These notions of observation functions are strictly equivalent (as discussed by Kaelbling et al. [1], footnote 4). Furthermore, in the foundational work of PSRs [2], observations were formulated to be emitted from the state left as well.
>
>
> > Does it make sense to start the computation from the stationary distribution
>  (that you do not have access to since you can't observe the states)? Moreover, there is a stark difference between beliefs that you infer from sequences of actions and observations and the underlying occupancy measure of the (hidden) MDP. Could you elaborate on that?
>
> A very interesting question! As you correctly point out, the fact we learn $b_\pi$ as the prior is a consequence of our exploration procedure. There are multiple ways you can look at this question:
>
> 1. Should the agent learn a system via uniform random exploration and immediately 'forget' the entire interaction history and reset the internal POMDP it has learned, then, by our ergodicity assumption, the distribution of states over the system is $b_\pi$ with high probability.
> 2. If an agent learns a system via our algorithm, leaves, and returns the next day, the agent has no control over whether the system has been manipulated by unseen forces. Prior mispecification, in practice, happens often.
> 3. Theoretical studies on related uncertain decision-making problems (Bayesian bandits, etc.) show that performance degrades 'gracefully' when a prior differs from the actual distribution of the problem [3], though actual verification of this fact for our setting would be the subject of future work.
>
> > Is the theory expandable to non-uniform exploration strategies, e.g., with stochastic behavioral policies having full support (as standard in regularized MDPs)?
>
> Yes! As long as the exploration policy is memoryless and the Markov chain $(a_t, s_t, o_t)$ is ergodic, then Theorem 1 still holds in the more general case. Thank you for the question -- we have updated the paper to state this point in a footnote.
>
>
> **References**
>
> [1] L. P. Kaelbling, M. L. Littman, and A. R. Cassandra, “Planning and acting in partially observable stochastic domains,” Artificial Intelligence, vol. 101, no. 1, pp. 99–134, May 1998.
>
> [2] S. Singh, M. James, and M. Rudary, “Predictive State Representations: A New Theory for Modeling Dynamical Systems,” UAI, 2004.
>
> [3] M. Simchowitz et al., “Bayesian decision-making under misspecified priors with applications to meta-learning,” in Advances in Neural Information Processing Systems, 2021.

---

### Official Review · Reviewer_atUB · 2025-10-31

**Soundness:** 2
**Presentation:** 2
**Contribution:** 2
**Rating:** 2
**Confidence:** 4

**Summary:**

The paper looks at how you infer a model of a POMDP through only sequences of actions and observations.  In essence, the goal is to recover a notion of state with corresponding transition and observation dynamics.  The work builds off of PSR representations and seeks to learn a transformation of the representation to recover the POMDP.  The works make assumptions about the POMDP and is only able to recover models up to a (fairly restrictive) observability partition.

**Strengths:**

The paper is exploring an interesting problem, which is certainly at the heart of representation learning.  This problem was particularly of interest 10-15 years ago, and to my knowledge, was not solved.  While PSRs have been shown to be able to be efficiently and directly estimated from an action-observation sequences, POMDPs have not.  The PSR model class is strictly largely than POMDPs, but they do no admit the same finite latent state interpretation.  Even if you knew the underlying environment could be modeled as a POMDP, we don't have direct methods to extract that POMDP.  State of the art (disappointingly)  is still likely EM.  Returning to an open problem that the community has "forgotten" about is valuable.

The paper takes a novel approach of extracting a "similarity transformation" of the latent state vector representation of a linear PSR to recover the POMDP states, observation, and transitions.

**Weaknesses:**

The paper has a number of weaknesses cutting across soundness, presentation, and contribution.

As for contributions, while it seeks to address an unsolved problem that was never solved, it doesn't really succeed at doing so.  The authors' notion of an observability partition, and Theorem 1's limitation to only matching the POMDP in aggregate over the partition seems to be dodging the open question of identifying a POMDP representation.  POMDPs whose states can identified from their next observation distribution is an extremely limiting class.  While the toyest of toy POMDPs might satisfy this restriction (e.g., any POMDP with <= 2 states must satisfy it, which includes Tiger), that doesn't make it an interesting class.  T-Maze, for example, does not have this property as the long hallway would exactly be collapsed by the observability partition.  And indeed, the experiments include only a truncated T-Maze (for which no description is given in the paper that I could find).  If we're unconcerned with the recovered POMDP allowing negative probabilities for states within a partition, then why not be happy with linear PSRs?  Furthermore, the abstract claims it produces "proper probabilistic models leveragable by downstream inference operations."  But they are not "proper probabilistic models".  And what downstream operations are being referred to, when none are given (besides planning, for which PSRs suffice and are performative by the paper's own experiments).

As for soundness, there are a number of statements that lack precision.
* The abstract claims "[PSRs] cannot, however, yield direct estimates of transition and observation likelihoods", which seems wrong as PSRs have simple closed form transition updates, observation distributions.  Indeed on line 178 the authors say, "A PSR can be used to compute the likelihood of observations ..."
* The Singh et al. (2004) paper defines a "system dynamics matrix", but the SDM is not a Hankel matrix at all (which 3.1 seems to suggest it is).  The first column of the SDM (assuming tests start with the empty test) would be all ones.  The first row of the SDM (assuming histories start with the empty history) would not be all ones, as some pairs of tests are mutually exclusive.
* Similarly, Equation (6) doesn't come from Wolfe nor is it the ${\hat P_{\cal T, H}}$ matrix from Boots.  Those both use the SDM.
* The paper makes an early remark that data is gathered with a uniform random exploration, but doesn't elaborate further or include it formally as an assumption.  Yet, equations (1-6) all depend on that requirement for them to make sense, otherwise conditioning on future actions might leak information about past observations.  See Bowling et al. (2006, ICML).
* Line 163: Doesn't this need proof or a citation?
* In 3.3, it sure seems like assumption 3 is redundant.  Assumption 2 states ${\bf Back}$ is full rank.  So, the rows of ${\bf Back}$ must be linearly independent, as the rows are the $|S|$ dimension in the $|S|\times\infty$.

In addition, the experiments are not particularly compelling.
* There is no description of what the shading in Figure 3 means.

**Questions:**

1. In what way does this method provide a "proper probabilistic model" that PSRs do not?
2. What would happen if the underlying system was a PSR that could not be represented with a finite-state POMDP?  What would your learning algorithm extract for the similarity transformation?

---

> ### Author Response · Authors · 2025-12-03
> **Response to atUB (part 1/2)**
>
> Dear reviewer,
>
> Thank you for taking the time to provide pointed feedback on our paper.
>
> >  The authors' notion of an observability partition, and Theorem 1's limitation to only matching the POMDP in aggregate over the partition seems to be dodging the open question of identifying a POMDP representation. POMDPs whose states can identified from their next observation distribution is an extremely limiting class.
>
> We agree that the test environments included in Section 5 (or the running example in Section 4) of the submitted paper do not highlight why the class of POMDPs our approach is capable of learning exhibit nontrivial partial observability. Indeed, all examples included in the submitted paper that are fully learnable have a unique max-likelihood observation that is uniquely emitted from that state.
>
> As mentioned in the shared response to all reviewers, we have clarified the language in the paper to state the class of learnable POMDPs more clearly in Section 4.1.1, which is a class broader than our original examples. We also include two additional POMDPs that have fewer observations than the number of states to have examples of POMDPs of that more general class. These environments require multiple observations to infer high probability in any given state when planning in section 5.
>
> > And indeed, the experiments include only a truncated T-Maze (for which no description is given in the paper that I could find).
>
> We have included descriptions of all domains used in experiments in Appendix C.5.
>
> >  If we're unconcerned with the recovered POMDP allowing negative probabilities for states within a partition, then why not be happy with linear PSRs?... And what downstream operations are being referred to, when none are given (besides planning, for which PSRs suffice and are performative by the paper's own experiments).
>
>
>
> Indeed, for the purposes of planning, PSRs and POMDPs are functionally equivalent. We agree that our experiments so far did not demonstrate why one might attempt to learn a POMDP _after_ an initial computation of a linear PSR.
>
> To help readers decide which model they would like to learn, we included a reward specification experiment inspired by our motivation to enable robots to learn and act in partially-observable systems. This experiment, described in section 5 (as mentioned in the shared response to all reviewers), involves (1) having the agent explore and learn a system *without* rewards and (2) finding a way to specify a reward to drive the system into a state with a certain observation distribution after the system has been learned. When learning a POMDP, we can design algorithms that can analyze the likelihoods of transition/observations learned to determine this reward. In the past, Boots et al. specified PSR rewards by rewarding individual observations [1]. Our experiments include an example where the latter may not elicit the correct behavior from the planner, but the former will.
>
> > Furthermore, the abstract claims it produces "proper probabilistic models leveragable by downstream inference operations."
>
> Thank you for pointing out that "proper probabilistic models" is a poor choice of phrasing, which we have removed in the new iteration of the draft. Indeed, since both PSRs and POMDPs correctly evaluate the likelihood of a test in a specific belief state, they are both 'probabilistic' in their own right.
>
> Instead, we have shifted our language to emphasize that our method reveals some notion of transition and observation likelihoods, even if they are not the full system dynamics. Our subsequent experiments show that we can still learn some interesting POMDPs despite this limitation and that learning explicit observation and transition likelihoods is useful.
>
> > The abstract claims "[PSRs] cannot, however, yield direct estimates of transition and observation likelihoods", which seems wrong as PSRs have simple closed form transition updates, observation distributions. Indeed on line 178 the authors say, "A PSR can be used to compute the likelihood of observations ..."
>
> We agree that linear PSRs have closed-form transitions and can recover observation distributions given the PSR's sufficient statistic. We emphasize that our algorithm also returns likelihoods of internal transitions and observations as conditioned on individual states. We have revised the language of our abstract to reflect this point.

---

> ### Author Response · Authors · 2025-12-03
> **Response to atUB  (part 2/2)**
>
> > The Singh et al. (2004) paper defines a "system dynamics matrix", but the SDM is not a Hankel matrix at all (which 3.1 seems to suggest it is). The first column of the SDM (assuming tests start with the empty test) would be all ones. The first row of the SDM (assuming histories start with the empty history) would not be all ones, as some pairs of tests are mutually exclusive.
>
> You are correct to point out that the SDM matrix is not the same as a Hankel matrix as referred to by Balle et al. [2]. They are closely related in the fact that rows of a Hankel matrix are scaled by a constant (e.g., renormalized to account for conditioning on histories after handling divide-by-zero issues, etc.), so they can be constructed in similar ways [3]. We cited Singh et al. 2004 because we found their construction elegant but modified the method for our (Hankel) purposes. We have also revised the language in Section 3 to include the distinction between the SDM and a Hankel matrix.
>
> > Similarly, Equation (6) doesn't come from Wolfe nor is it the...
>  ...matrix from Boots. Those both use the SDM.
>
> We agree that the equation is reminiscent of the way that SDMs are estimated. The citation was due to the subsequence-counting technique advocated by both Wolfe et al. and Boots et al. (e.g., the history-suffix algorithm). In Equation 6, the denominator is taking frequency counts based on actions in the sequence and not entire histories, as is done when estimating an SDM.
>
> > The paper makes an early remark that data is gathered with a uniform random exploration, but doesn't elaborate further or include it formally as an assumption. Yet, equations (1-6) all depend on that requirement for them to make sense, otherwise conditioning on future actions might leak information about past observations. See Bowling et al. (2006, ICML).
>
> Thank you for the reference and astute observation! We have included a brief description of how equations (1)-(6) relate to our assumptions and that, in general, the exploration procedure must be stationary and memoryless. We have added a citation to the included work and other work that requires similar memorylessness assumptions on the exploration policy.
>
> > Line 163: Doesn't this need proof or a citation?
>
> We have proved this claim as a step in our proof of Proposition 1, included in Appendix A. We have added a phrase to make this clear.
>
> > In 3.3, it sure seems like assumption 3 is redundant. Assumption 2 states
>  is full rank...
>
> Please see the shared response to all reviewers on the discussion of assumptions 3 and 4. We have improved our algorithm slightly in Section 4.3, removing the need for assumption 4.
>
> > There is no description of what the shading in Figure 3 means.
>
> Thank you for pointing that out. The caption of Figure 3 and all other figures has been updated to say what the error bars/shading/bounds mean (usually, standard deviation).
>
> > What would happen if the underlying system was a PSR that could not be represented with a finite-state POMDP? What would your learning algorithm extract for the similarity transformation?
>
> Our guarantee (Theorem 1) only holds if the data collected by the agent was generated by a finite-state POMDP. Even if we had *linear* PSR that had an associated SDM or Hankel matrix with finite rank, we may not have the diagonal-structured observation matrices that can be exploited by an eigenvalue problem to recover a similarity transform.
>
>
> **References**
>
> [1] B. Boots, S. M. Siddiqi, and G. J. Gordon, “Closing the learning-planning loop with predictive state representations,” The International Journal of Robotics Research, 2011.
>
> [2] B. Balle, X. Carreras, F. M. Luque, and A. Quattoni, “Spectral learning of weighted automata,” Mach Learning, 2014.
>
> [3] P.-L. Bacon, B. Balle, and D. Precup, “Learning and Planning with Timing Information in Markov Decision Processes,” UAI, 2015.

---

### Official Review · Reviewer_NVMq · 2025-11-01

**Soundness:** 3
**Presentation:** 2
**Contribution:** 3
**Rating:** 6
**Confidence:** 3

**Summary:**

This paper addresses the problem of learning discrete POMDP parameters from action-observation sequences without prior knowledge of the state space or transition dynamics. The authors make a solid theoretical contribution by establishing a novel connection between Predictive State Representations (PSRs) and tensor decomposition methods for POMDP learning. Their key result (Theorem 1) characterizes what can be learned from observations: the method recovers transition and observation matrices up to a full-rank observability partition, where states sharing identical observation distributions across all full-rank actions are grouped together. This relaxes restrictive assumptions from prior tensor-based approaches that required unique observations per state for each action. The authors develop a theory-driven algorithm that: (1) constructs a Hankel matrix from data, (2) performs spectral decomposition to obtain a PSR model, (3) recovers the similarity transform using joint diagonalization of observation matrices from full-rank actions, and (4) extracts probabilistic POMDP parameters. Experiments on small-scale benchmarks (Tiger, T-Maze, Sense-Float-Reset with 2-4 states) demonstrate convergence of learned parameters to ground truth and show that planning performance matches that of the true POMDP model, validating the theoretical framework's practical viability.

**Strengths:**

**Solid theoretical contribution**: Establishes the first formal connection between PSRs and tensor decomposition methods for POMDP learning. Theorem 1 characterizes learnability—recovering transitions/observations up to full-rank observability partitions—and relaxes prior assumptions by requiring observation uniqueness only across all full-rank actions (not per-action), enabling learning of a broader class of POMDPs.

**Principled theory-driven algorithm**: Clear pipeline from Hankel matrix construction to PSR learning to similarity transform recovery via joint diagonalization. The approach of using full-rank actions as anchors to extract probabilistic parameters is elegant and directly motivated by theory.

**Convincing experimental validation**: Learned parameters converge to ground truth with <10⁵ samples for small problems, and planning performance matches true POMDP models. Figure 4 visualization provides clear evidence of meaningful recovery.

**Well-motivated problem**: Addresses fundamental gap where PSRs lack explicit probabilities for inference while tensor methods are too restrictive. Real-world examples (hidden locking mechanisms) effectively motivate the work.

**Weaknesses:**

**Major:**

- **Limited experimental scale and analysis**: Experiments only test 2-4 state problems with no evaluation of how performance scales with state space size. Runtime complexity O(|S|(|A||O|)^(2n_obs+2)) is exponential in observability length, but no empirical analysis demonstrates where this becomes prohibitive. Missing comparisons with other POMDP learning methods (e.g., EM-based approaches, other spectral methods). While the experiments serve to validate the theory, these limitations significantly reduce the practical contribution of the algorithm. A clearer failure analysis showing where the method breaks down and computational time comparisons would strengthen the work.
- **Restrictive algorithmic requirements reduce practical applicability**: The algorithm fundamentally requires at least one action with a full-rank transition matrix to recover observations—many real-world POMDPs may lack this property (e.g., deterministic systems, heavy aliasing). The ergodicity assumption may not hold in systems with absorbing states or disconnected components. While the paper is honest about partition-level recovery, it underemphasizes how these structural requirements limit the class of learnable POMDPs in practice.
- **Together, these limitations constrain practical impact**: The combination of scalability concerns and structural assumptions means the algorithm's applicability remains narrow, despite the solid theoretical contribution.

**Minor:**

- **Presentation could be clearer**: Dense mathematical exposition without pseudocode makes the algorithm difficult to follow. An intuitive explanation before diving into formalism and explicit algorithm pseudocode would improve accessibility.
- **Missing ablation studies**: No experiments examining key design choices (e.g., impact of normalization step in Section 4.3, joint vs. per-action diagonalization, sensitivity to Hankel matrix size).
- **Limited engagement with recent work**: Would benefit from discussion of or comparison with recent neural approaches to POMDP learning and connections to related system identification literature, though this is not critical for the theoretical contribution.

**Note**: These weaknesses do not diminish the solid theoretical contribution, but they do limit the overall impact of the work.

**Questions:**

**Scalability**: Can the authors provide experiments or analysis on larger state spaces (e.g., 10-20 states) to demonstrate where the method becomes computationally infeasible?

**Full-rank action requirement**: How restrictive is requiring at least one full-rank action in practice? Can the authors provide guidance for practitioners to verify this assumption holds for their domain, and characterize what happens empirically when violated?

**Hyperparameter sensitivity**: Table 1 shows widely varying parameters across domains (e.g., 1/κ from 0.015 to 0.34). How sensitive is the algorithm to these choices? Can the authors provide principled selection guidance or demonstrate robustness through sensitivity analysis?

---

> ### Author Response · Authors · 2025-12-03
> **Reply to NVMq**
>
> Dear reviewer,
>
> Thank you for taking the time to thoroughly engage with our work! We appreciate the recognition of the theoretical contribution of our paper. We have addressed many of your main points in the shared response and your remaining points below:
>
> We address each of these points together:
> > Limited experimental scale and analysis: Experiments only test 2-4 state problems with no evaluation of how performance scales with state space size. Runtime complexity O(|S|(|A||O|)^(2n_obs+2)) is exponential in observability length, but no empirical analysis demonstrates where this becomes prohibitive.
>
> > A clearer failure analysis showing where the method breaks down and computational time comparisons would strengthen the work.
>
> > Scalability: Can the authors provide experiments or analysis on larger state spaces (e.g., 10-20 states) to demonstrate where the method becomes computationally infeasible?
>
> > Hyperparameter sensitivity: Table 1 shows widely varying parameters across domains (e.g., 1/κ from 0.015 to 0.34). How sensitive is the algorithm to these choices? Can the authors provide principled selection guidance or demonstrate robustness through sensitivity analysis?
>
> > Missing ablation studies: No experiments examining key design choices (e.g., impact of normalization step in Section 4.3, joint vs. per-action diagonalization, sensitivity to Hankel matrix size).
>
> To address the first four points, we have included experiments on larger-scale domains (4-14 states) with runtime information and also results under various Hankel matrix sizes and choices of $1 / \kappa$ in Appendix C.2., with some discussion on how to choose parameters in practice. The analysis suggests that it is preferable to estimate Hankel matrices as *large* as possible (as accommodated by memory and available runtime) to ease the choice of the remaining parameters of the algorithm.
>
> Each step (joint eigendecomposition and normalization) of the algorithm is critical for Theorem 1 to hold. Without them, the algorithm will return invalid output for POMDPs that have individual actions with redundant observations, etc., which all the experimental domains used in our paper contain. We do not believe ablating these steps would yield any additional insight than that which is already suggested by our theoretical analysis.
>
> We address these points together:
> > The algorithm fundamentally requires at least one action with a full-rank transition matrix to recover observations—many real-world POMDPs may lack this property (e.g., deterministic systems, heavy aliasing). The ergodicity assumption may not hold in systems with absorbing states or disconnected components. While the paper is honest about partition-level recovery, it underemphasizes how these structural requirements limit the class of learnable POMDPs in practice.
>
> > Full-rank action requirement: How restrictive is requiring at least one full-rank action in practice? Can the authors provide guidance for practitioners to verify this assumption holds for their domain, and characterize what happens empirically when violated?
>
> As stated in the shared response to all reviewers, we have included a discussion of the restrictiveness of the assumptions made by our algorithm in Section 4.3.3.
>
>
> > Presentation could be clearer: Dense mathematical exposition without pseudocode makes the algorithm difficult to follow.
>
> As mentioned in the shared response to all reviewers, we have attempted to simplify components in the algorithm, which leads to a simpler exposition overall.  We have also included algorithm pseudocode in Appendix B.1.
>
> > Limited engagement with recent work: Would benefit from discussion of or comparison with recent neural approaches to POMDP learning and connections to related system identification literature, though this is not critical for the theoretical contribution.
>
> We have updated the Related Works (Section 6) section with a more robust discussion of related neural approaches to POMDP learning and system identification literature.

---

### Author Response · Authors · 2025-12-03
**Shared Response to All Reviewers**

To all reviewers,

Thank you for taking the time to review our paper and providing thorough feedback. There were many repeated themes across all four reviews, which we address in this comment. Feedback not shared among reviewers is addressed in individual replies. We have also included a latexdiff document in the supplementary details so that all document changes are easy to find.


**Discussion of restrictiveness of assumptions and POMDPs learnable by our method.**

We realize that our paper was not clear on the types of POMDPs our algorithm can learn, and some reviewers (reasonably) interpreted our result as only being able to learn POMDPs that consist of observations that fully-identify the state of the system. Our algorithm can learn a more general class of POMDPs where the observation distributions across full-rank actions are *distinct*. This includes POMDPs that have fewer observations than states, which requires observing multiple observations to correctly infer the underlying state. We have introduced two new experimental domains in Section 5, "noisy hallway" and "directional hallway" (full details in Appendix C.5.3) that have this feature.

We have added a sentence to the Assumption section (3.3) to discuss the assumption of full-rank forward and back matrices. We have also added a new section 4.1.1 to discuss the restrictiveness of the assumptions/presence of full-rank actions.

**On the value of learning POMDPs**
Another concern raised by reviewers was that our algorithm does not yield interpretable likelihoods *within* observability partitions, which does not give our technique any advantages relative to PSRs, especially if they yield the same planning performance. We clarify first that we can learn a class of nontrivial POMDPs fully (as stated above). We also introduce an additional experiment (Section 5, "Planning performance on specified rewards.") that highlights the advantage of learning POMDPs, in that we can learn the dynamics and observation models of the world _once_ and modify the reward function and immediately replan, rather than learning a new representation.


**Baselines**: We have added EM baseline to this paper (Section 5, Figures 3 and 4). Furthermore, the reward-specification experiment introduced in Section 5 suggests limitations of implicit latent-represent type methods, which include modern approaches that use recurrent neural architectures (RNNs, etc.).

**Scalability of methods to larger POMDPs**: The scalability of our method to larger POMDPs is indeed a weakness of the specific algorithm described in our paper. Extending the scale of our method is critical to making our technique broadly relevant.  Our investigations show that there are likely techniques that will scale, but are sufficiently complex that they must be outside the scope of the first investigation described in this paper. We emphasize that the subject of *this* paper is to learn a broader class of POMDPs than we could with previous methods and we have added experiments in Section 5 that we hope highlight the value of learning these kinds of models. Existing learning methods could not even learn simple toy problems like Tiger [1]. Future research into scaling our method would focus on incrementally learning specific *structure* in the POMDP to enable scaling - for example, by factoring large POMDPs into smaller ones, as noted in the conclusion of this paper.




**Additional Improvements**:

- We found a simpler way to perform the 'transition normalization' step of the algorithm, which removes the need for the fourth assumption (linear independence of observations distributions) in Section 3.3. That assumption was required so that we could take the matrix the Moore-Penrose inverse of $\mathbf{Back}_{obs}$. Since we now have a new procedure only involving the 'final vector' of the linear PSR, we may omit this assumption.

- We added descriptions of each of the evaluation environments in the appendix.

- We fixed a bug in our PO-UCT planner, yielding stronger planning performance in Section 5, Figure 3.

**References**

[1] K. Azizzadenesheli, A. Lazaric, and A. Anandkumar, “Reinforcement Learning of POMDPs using Spectral Methods,” in Conference on Learning Theory, PMLR, June 2016, pp. 193–256.

---

### Meta-Review · Area_Chair_ZUAv · 2026-01-15

**Summary:**

The paper introduces a spectral method to learn POMDP parameters (transitions, observations) from action-observation sequences by recovering a similarity transform from a learned Linear PSR. This approach recovers parameters up to an "observability partition". Reviewers praised the theoretical novelty of linking PSRs to explicit POMDP recovery. However, significant concerns were raised regarding the restriction to toy environments (2-4 states), strong assumptions (ergodicity, full-rank actions), and the practical utility of the method compared to PSRs or EM-based baselines. Unfortunately, I am advocating (weakly) for rejection because I do not believe the most important criticisms surrounding the practical utility of "observability partitions" and the applicability of this proposed method on realistic environments with large state spaces have been adequately addressed.

**Reviewer Concerns:**

Addressed by Rebuttal:

- Baselines (Reviewers NVMq, MsTT): The authors added an Expectation-Maximization (EM) baseline, addressing the lack of comparative evaluation.

- Scope of Learnable POMDPs (Reviewers NVMq, atUB): The authors clarified that the method is not limited to states with unique observations, but rather states with distinct observation distributions across full-rank actions. They added "noisy hallway" and "directional hallway" experiments to demonstrate this capability.

- Algorithm Clarity & Assumptions (Reviewers atUB, MsTT): The authors simplified the algorithm (removing Assumption 4), added pseudocode, and included a dedicated section discussing the restrictiveness of assumptions. They also corrected imprecise claims regarding "proper probabilistic models."

- Utility of POMDPs vs. PSRs (Reviewer atUB): The authors added a reward-specification experiment to argue that recovering explicit POMDP parameters allows for re-planning with new rewards without re-learning dynamics, a capability lacking in pure PSRs or RNNs.

Outstanding:

- Scalability (Reviewers NVMq, iKCN, MsTT): While the authors increased the experiment scale to 14 states, the method's complexity remains exponential in observability length. The approach is fundamentally limited to small, discrete state spaces compared to modern deep RL benchmarks.

- Practical Applicability (Reviewer atUB, iKCN): The requirement for full-rank actions and ergodicity under uniform exploration is a hard constraint that likely precludes this method from being applied to many real-world partial observability problems.

**Reviewer Scores:**

Reviewer NVMq (6 -> 6): The reviewer was already positive about the theory. The addition of baselines and runtime analysis addresses their main critiques.

Reviewer atUB (2 -> 4): This reviewer was highly critical of the contribution's significance. The authors' clarification that the method handles non-trivial partial observability (beyond unique observations) and the removal of the term "proper probabilistic models" addresses specific technical objections. However, skepticism regarding the practical utility of "observability partitions" likely remains.

Reviewer MsTT (6 -> 6): The reviewer provided constructive feedback which the authors largely implemented (baselines, assumption discussion).

Reviewer iKCN (4 -> 4): The scale increase (up to 14 states) improves the optics of the evaluation, though the fundamental skepticism regarding the method's utility versus model-free alternatives likely prevents a higher score.

---

### Decision · Program_Chairs · 2026-01-26

Reject